

# Modeling seasonal variations of extreme rainfall on different time scales in Germany

Jana Ulrich[1], Felix S. Fauer[1], and Henning W. Rust[1]

[1]Institute of Meteorology, Freie Universität Berlin, Carl-Heinrich-Becker-Weg 6-12, 12165 Berlin, Germany

**Correspondence:** Jana Ulrich (jana.ulrich@met.fu-berlin.de)

**Abstract.** We model monthly precipitation maxima at 132 stations in Germany for a wide range of durations from one minute to about six days using a duration-dependent generalized extreme value (d-GEV) distribution with monthly varying parameters. This allows for the estimation of both monthly and annual intensity–duration–frequency (IDF) curves: (1) The monthly IDF curves are steeper in summer and exhibit higher intensities for short durations than in the rest of the year. Thus, everywhere in Germany short convective extreme events occur very likely in summer. In contrast, extreme events with a duration of several hours up to about one day are more likely to occur within a longer period or even spread throughout the whole year, depending on the station. There are major differences within Germany with respect to the months in which long-lasting stratiform extreme events are more likely to occur. At some stations the IDF curves (for a given quantile) for different months intersect. The meteorological interpretation of this intersection is that the season at which a certain extreme event is most likely to occur shifts from summer towards autumn or winter for longer durations. (2) We compare the annual IDF curves resulting from the monthly model with those estimated conventionally, that is, based on modeling annual maxima. We find that adding information in the form of smooth variations during the year leads to a considerable reduction of uncertainties. We additionally observe that at some stations, the annual IDF curves obtained by modeling monthly maxima deviate from the assumption of scale invariance, resulting in a flattening in the slope of the IDF curves for long durations.

## 1 Introduction

Extreme precipitation events can potentially cause significant damage (Linnerooth-Bayer and Amendola, 2003; Barredo, 2009; Davenport et al., 2021), depending on their duration and spatial extent: extreme convective events can lead to flash floods, while long-lasting stratiform precipitation may lead to river flooding. In recent years, floods and landslides following heavy precipitation have become increasingly frequent in many European countries (Bronstert, 2003; Paprotny et al., 2018), and weather conditions favoring the occurrence of heavy rainfall events are expected to further increase due to anthropogenic climate change (Hattermann et al., 2013; Hartmann et al., 2013, and references therein). However, in addition to regional differences, changes in the frequency and intensity of extreme precipitation in Europe have been found to also differ between different storm types, namely convective and stratiform events (Berg et al., 2013), as well as between different seasons (e.g., Moberg and Jones, 2005; Łupikasza, 2017; Kunz et al., 2017, and references therein). Hence, it is critical to research and





improve our understanding of the occurrence of extreme precipitation events on different time scales as well as in different seasons.

The characteristics of extreme precipitation on different time scales can be summarized in terms of intensity–duration–frequency (IDF) curves. These are a standard tool in hydrology for designing hydrological structures and managing water supplies (Durrans, 2010). IDF curves are basically probability distributions for extreme values of precipitation intensity for

a range of durations, or more precisely aggregation times. Thus, they provide the relationship between precipitation intensity and duration for selected occurrence frequencies (i.e. exceedance probabilities or return periods). Since differences exist in the storm characteristics of different seasons, it is essential to provide information on precipitation extremes on a seasonal basis. Even though a seasonal resolution may not be relevant for planning or adjusting hydrological structures, it would be beneficial for stakeholders in agriculture or water storage. In addition, a seasonal approach allows for a more detailed examination of

the underlying mechanisms that influence the IDF relationship. However, while studies exist that investigate seasonality in extreme precipitation of a selected duration (e.g., Maraun et al., 2009; Rust et al., 2009; Fischer et al., 2018, 2019) or also in flood frequency (e.g., Durrans et al., 2003; Kochanek et al., 2012; Rottler et al., 2020), there are few studies regarding seasonal IDF curves (Willems, 2000; Durrans, 2010).

Extreme value theory offers several approaches to describe the occurrence probability of extreme events (for an introduction,

see Coles, 2001). There are numerous applications of extreme value statistics in hydrology and climatology (e.g., Katz et al., 2002; Friederichs, 2010; Davison and Gholamrezaee, 2012; Papalexiou and Koutsoyiannis, 2013; Sebille et al., 2017; Lazoglou et al., 2019), among which the block maxima approach is a commonly accepted concept. For this approach, the observed time series is divided into blocks of equal length and the probability distribution of the maxima of these blocks is modeled using a generalized extreme value (GEV) distribution. Since extreme events are by definition rare, the estimation of quantiles (return

levels) corresponding to small exceedance probabilities (return periods) is always associated with the problem of limited data. When modeling IDF curves, the limitations of the observations are firstly the spatial coverage and secondly the temporal resolution (Courty et al., 2019). The German Meteorological Service (DWD) operates a relatively dense station network, so that many long observation time series exist for daily precipitation measurements. However, fewer stations provide sub-daily measurements and, in addition, considerably shorter time series are available at these stations, since operating instruments with

hourly or minutely measurement intervals has only been feasible without considerable maintenance for a few decades. The objective in modeling sub-daily extreme precipitation events is therefore to use the available data most efficiently, i.e. to pool the information where possible. Hence, in this study we aim to combine different information on extreme precipitation within one model, namely information on different durations as well as seasonal variations.

In order to assess extreme precipitation observations of different aggregation times simultaneously, it is possible to use a

duration-dependent extreme value distribution (Koutsoyiannis et al., 1998; Van de Vyver and Demarée, 2010; Lehmann et al., 2013; Van de Vyver, 2018). In the context of the block maxima approach, a duration-dependent GEV (d-GEV) distribution is derived by implementing empirical dependencies of the GEV parameters on duration. Thus, we are able to directly obtain quantile estimates for all durations within the considered interval while additionally reducing the uncertainties of the estimation by combining information of different durations (Ulrich et al., 2020). To our knowledge, this approach has so far only been





used with an annual block size. This means that only the annual maxima of each aggregation time are used, and therefore large amounts of data are neglected for the analysis. However, when modeling daily precipitation sums, monthly block sizes have been shown to be sufficient to model extreme precipitation in the mid-latitudes (Coles, 2001; Maraun et al., 2009; Rust et al., 2009). Although, the choice of a monthly block size requires a more complex model that explicitly includes the intra-annual variations. This can be accomplished by adding smooth periodic functions as covariates for the GEV parameters (Fischer et al.,

2018, 2019). Fischer et al. (2018) demonstrated that this approach provides more precise quantile estimates as it allows the use of more data.

In this study, we implement monthly covariates analogously for the parameters of the d-GEV distribution. Hence, we model intra-annual variations of extreme precipitation for a wide range of durations from one minute to approximately 6 days at 132 stations in Germany. This not only allows us to estimate and compare IDF curves of different months, but we also expect to

obtain more reliable annual IDF curves due to the more efficient use of the data. This study addresses the following research questions:

– How does the IDF relationship at different stations in Germany evolve throughout the year?

– To what extent do the annual IDF curves based on monthly and annual maxima differ?

– Does modeling monthly maxima allow us to infer the dependence of distribution parameters on duration?

The remainder of this study is organized as follows: In Sect. 2, we present the data and methods on which this study is based. We address both the methods used for modeling as well as for comparing the different models. We then present the respective results regarding our research questions in Sect. 3 and discuss them in Sect. 4. We close with our conclusions in Sect. 5.

## 2  Methods

We aim to model the intra-annual variations of extreme precipitation on different time scales. For this purpose, we use obser-

vations with high temporal resolution from stations in Germany. We use a duration-dependent GEV (d-GEV) distribution with monthly covariates to describe the monthly maxima over a range of durations collectively in one model. Thereby, appropriate models for the intra-annual variations of the d-GEV parameters are selected through step-wise forward regression. This approach allows us to examine how the IDF curves vary throughout the year in different areas of Germany. From this seasonal model, we can derive annual IDF curves as well. We compare these annual IDF curves with those resulting from directly

modeling the annual maxima via a verification procedure using the quantile skill index. Finally, we verify whether modeling monthly maxima allows for a more precise estimate of the relationships between GEV parameters and duration. Therefore, we model each duration separately using the GEV distribution with monthly covariates. Details of the data as well as all methods involved are described in the following section.

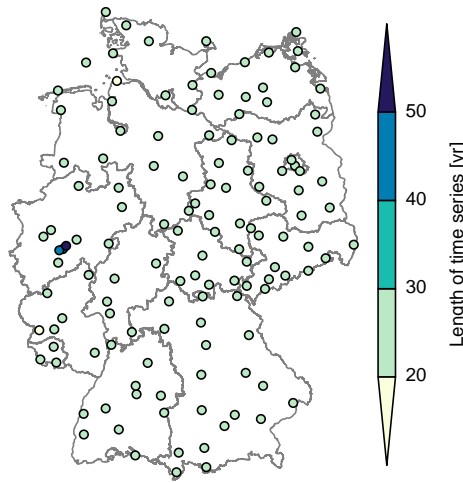

**Figure 1.** Map of Germany with positions of all 132 stations considered. Colors represent the length of the available time series with minute resolution. The longest observation period of 51 years exist for station Bever-Talsperre (dark blue).

## 2.1 Data

We use precipitation measurements at 132 stations in Germany that provide a temporal resolution of one minute. Their locations are presented in Fig. 1. The majority (129) of these stations are operated by the German Meteorological Service (DWD)[1]. The available time series at these stations range from 19 to 28 years (Fig. 1 yellow and green). Additionally we use three stations operated by the Wupperverband[2] with time series $\geq 44$ years (Fig. 1 blue). The station Bever-Talsperre with the longest observation period of 51 years is used as example station.

The observations were accumulated to the following durations: $d \in 2^{\{0,1,2,..,13\}} \min = \{1, 2, 4, ..., 8192\}$ min, with the longest duration $8192 \min \approx 5.7$ days. Thus, resulting in 14 time series per station. Of each time series, we consider both the monthly and annual maxima. Blocks are excluded from the analysis if they contain more than 10% missing values.

## 2.2 Modeling annual maxima of different durations

The challenge in modeling extremes is to estimate probabilities of very rare or even not yet observed events. Here, we apply
the block maxima approach which is commonly used for this purpose. It is based on the Fisher–Tippett–Gnedenko Theorem, which essentially states that under certain assumptions the probability distribution of block maxima can be modeled by the generalized extreme value (GEV) distribution (Coles, 2001).

---

[1]The data were obtained via ftp://ftp-cdc.dwd.de/climate_environment/CDC/observations_germany/climate.
[2]https://www.wupperverband.de





More precisely, let $X_1, ... X_n$ be a sequence of $n$ random variables which are independent and identically distributed (iid), with an unknown distribution. We denote the maximum of this sequence as

$$M_n = \max\{X_1, ..., X_n\}. \tag{1}$$

In the limit of large block sizes $n$, the non-exceedance probability can be approximated by the generalized extreme value (GEV) distribution

$$\Pr\{M_n \leq z\} \approx G(z), \tag{2}$$

if for $n \to \infty$ the distribution of properly rescaled $M_n$ converges to a non-degenerate distribution. The GEV distribution

$$G(z; \mu, \sigma, \xi) = \exp\left\{-\left[1 + \xi\left(\frac{z - \mu}{\sigma}\right)\right]^{-1/\xi}\right\} \tag{3}$$

is defined on $\{z : 1 + \xi(z - \mu)/\sigma > 0\}$ and has three parameters: location parameter $-\infty < \mu < \infty$, scale parameter $\sigma > 0$, and a shape parameter $-\infty < \xi < \infty$. Thus, the position and width of the distribution are specified by $\mu$ and $\sigma$, respectively. Whereas $\xi$ determines the right tail behavior, resulting in bounded right tails for $\xi < 0$ and polynomial decay for $\xi > 0$. In the case $\xi = 0$ Eq. (3) is interpreted in the limit of $\xi \to 0$, leading to the Gumbel distribution, with an exponentially decaying tail.

The GEV distribution is thus likely to be a well suited model for the distribution of annual precipitation intensity maxima of one selected aggregation duration. In order to model the distribution for different durations simultaneously, Koutsoyiannis et al. (1998) proposed that the empirical relationship between precipitation intensity and duration can be directly used to model the parameters of the GEV distribution depending on duration, which leads to a duration-dependent GEV (d-GEV) distribution $G(z, d; \mu(d), \sigma(d), \xi(d))$. The relationship between precipitation intensity $I$ and duration $d$ for a chosen non-exceedance probability $p$ corresponds to the quantile $q_p(d)$ of the d-GEV distribution:

$$I_p(d) = q_p(d) = \mu(d) - \frac{\sigma(d)}{\xi(d)}\left[1 - \{-\log(1 - p)\}^{-\xi(p)}\right]; \tag{4}$$

hence IDF curves can be estimated in a consistent way (Ulrich et al., 2020). For the empirical dependence of the parameters on duration, we follow the assumptions of Koutsoyiannis et al. (1998):

$$\sigma(d) = \sigma_0\left(\frac{d}{1\,\mathrm{h}} + \theta\right)^{-\eta}, \tag{5}$$

$$\mu(d) = \tilde{\mu} \cdot \sigma(d), \tag{6}$$

$$\xi(d) = \mathrm{const.}, \tag{7}$$

with re-parameterized location parameter $-\infty < \tilde{\mu} < \infty$, scale offset $\sigma_0 > 0$, duration offset $\theta \geq 0$ and duration exponent $0 < \eta \leq 1$. These assumptions are commonly used (Lehmann et al., 2013; Van de Vyver, 2015; Stephenson et al., 2016; Ritschel et al., 2017), however, it may be beneficial to introduce additional parameters (Van de Vyver, 2018; Fauer et al., 2021). By inserting assumptions (5)-(7) into Eq. (3), we obtain the d-GEV distribution with 5 parameters

$$G(z, d; \tilde{\mu}, \sigma_0, \xi, \theta, \eta) = \exp\left\{-\left[1 + \xi\left(\frac{z}{\sigma_0(d + \theta)^{-\eta}} - \tilde{\mu}\right)\right]^{-1/\xi}\right\}, \tag{8}$$





which constitutes a model for the distribution of annual precipitation maxima for a range of durations.

### 2.3 Modeling monthly maxima

According to the Fisher–Tippett–Gnedenko-Theorem, the GEV distribution is an adequate model for block maxima if the

block size is sufficiently large. For geophysical applications, such as modeling extreme precipitation, it is common to choose a block size of one year, as explicit modeling of seasonality is thereby avoided. However, this results in two major disadvantages: Large portions of the data are lost for the analysis if only the annual maxima are used; furthermore the assumption of identically distributed precipitation events can hardly be motivated if a pronounced annual cycle exists. Therefore, multiple studies suggest that a monthly block size is sufficient to model extremes of daily precipitation sums in the mid-latitudes (Coles, 2001; Maraun

et al., 2009; Rust et al., 2009; Fischer et al., 2018). Similarly, we use monthly maxima to model extreme precipitation of different durations: either with separate models for each duration using the GEV (Eq. (3)) or simultaneously by using the d-GEV distibution (Eq. (8)). Inspection of the quantile-quantile (q-q) plots indicates that the d-GEV distribution is a reasonable approximation for the distribution of monthly maxima at the regarded stations. The q-q plots for station Bever-Talsperre with respect to each month are shown in Fig. A1.

To account for any form of variability in the GEV model (Eq. (3)), the GEV parameters $\varphi \in \{\mu, \sigma, \xi\}$ can be modeled as linear functions of covariates $x_i$ within the framework of vector generalized linear models (VGLM) (Yee and Stephenson, 2007)

$$l^\varphi \left( \varphi(x_i) \right) = \varphi_0 + \sum_{i=1}^{I} \beta_i^\varphi \, x_i, \tag{9}$$

where $\varphi_0$ represents the intercept and $\beta_i^\varphi$ are the regression coefficients. The choice of the parameter specific link function $l^\varphi(\cdot)$

can ensure parameters to stay within a predefined range. However, we employ the identity $l^\varphi(\varphi) = \varphi$ as link function for all parameters. Following (Fischer et al., 2018), the intra-annual variations of the GEV parameters can be modeled as a periodic functions of the day of the year (doy) by using a series of harmonic functions with a fundamental period of one year:

$$\varphi(\text{doy}) = \varphi_0 + \sum_{j=1}^{J} \left[ \alpha_j^\varphi \cos \left( \frac{2\pi j \cdot \text{doy}}{365.25} \right) + \beta_j^\varphi \sin \left( \frac{2\pi j \cdot \text{doy}}{365.25} \right) \right], \tag{10}$$

where $J$ is the maximum order of harmonic functions. To obtain the parameters for each month, Eq. (10) is evaluated at the

corresponding center-days of each month. We model the seasonal variations of the d-GEV distribution in exactly the same way. Essentially, this means that each of the parameters $\varphi_{\text{d-GEV}} \in \{\tilde{\mu}, \sigma_0, \xi, \theta, \eta\}$ can be expressed in the form of Eq. (10).

### 2.4 Parameter estimation

The parameters of the GEV distribution can be estimated from a time series of observed block maxima. For this purpose, we apply the widely used maximum likelihood estimator (MLE) (Coles, 2001). Thus, the parameters are chosen by optimizing the





likelihood

$$\mathcal{L}(\phi \mid \mathbf{Z}) = \prod_{n \in N} g(z_n; \phi), \tag{11}$$

where the parameter vector $\phi = (\mu, \sigma, \xi)^T$ contains the unknown GEV parameters, the vector $\mathbf{Z}$ consists of the observed maxima $z_n$ for different blocks (years/ months) $n$ and $g(z_n; \phi)$ is the probability density function of the GEV distribution. This can be applied analogously for the d-GEV distribution:

$$\mathcal{L}(\phi \mid \mathbf{Z}) = \prod_{d \in D} \prod_{n \in N} g(z_{n,d}, d; \phi), \tag{12}$$

whereas in this case the parameter vector $\phi = (\tilde{\mu}, \sigma_0, \xi, \theta, \eta)^T$, $\mathbf{Z}$ now contains all observed maxima $z_n$ for different blocks (years/ months) $n$ and durations $d$ and $g(z_{n,d}, d; \phi)$ is the probability density function of the d-GEV distribution. A benefit of the MLE is that it can be easily extended in the case of using covariates to model the parameters (Eq. (10)). The parameter vector then contains the parameter intercepts $\varphi_0$ and regression coefficients $\alpha_j^\varphi$ and $\beta_j^\varphi$ for each parameter in the case of both

GEV and d-GEV distribution.

Since the the logarithm of the likelihood reaches the maximum at the same value, but is easier to calculate, the parameters are estimated by optimizing the log-likelihood numerically

$$\hat{\phi} = \arg\max_{\phi} \{\ln[\mathcal{L}(\phi \mid \mathbf{Z})]\}. \tag{13}$$

It is possible to derive the uncertainty of the parameter estimates, i. e. the variance-covariance matrix, via the Fisher information
matrix estimated in this process.

Nevertheless, Eqs. (11) and (12) are only valid if the block maxima are independent of each other. The assumption that maxima of different years or also months are independent is reasonable. However, a dependency exists between the maxima of different durations . Since Jurado et al. (2020) have shown that accounting for asymptotic dependence between durations yields limited improvement in estimating quantiles and comes at the cost of increased model complexity, we neglect it for
estimating the d-GEV parameters using Eq. (12). Yet, the dependence between durations is taken into account when estimating the uncertainties of the quantiles by using the bootstrap method (see Sect. 2.6).

### 2.5    Model selection

To obtain a parsimonious model, we use a selection procedure consisting of two steps: In the first step, we determine for which of the GEV/ d-GEV parameters the modeling of the intra-annual variations is not appropriate and which should therefore
remain constant. In the second step, we select which terms of the harmonic series in Eq. (10) are actually needed in order to model the non-constant parameters.

When modeling intra-annual variations of GEV parameters, the shape parameter $\xi$ is often assumed to be constant (Maraun et al., 2009; Rust et al., 2009; Fischer et al., 2018). This is justified by the fact that $\xi$ controls the tail of the distribution, and thus the estimation of $\xi$ is already associated with large uncertainties. Hence, adding additional coefficients to the estimation





**Table 1.** All possible models in the first step of the step-wise regression (GEV case) with $\omega = \frac{2\pi}{365.25}$

| $\mu$ | $\sigma$ | $\xi$ |
|---|---|---|
| $\mu_0$ | $\sigma_0$ | $\xi_0$ |
| $\mu_0 + \alpha_1^\mu \cos(\omega \cdot \text{doy})$ | $\sigma_0$ | $\xi_0$ |
| $\mu_0$ | $\sigma_0 + \alpha_1^\sigma \cos(\omega \cdot \text{doy})$ | $\xi_0$ |
| $\mu_0 + \alpha_1^\mu \cos(\omega \cdot \text{doy})$ | $\sigma_0 + \alpha_1^\sigma \cos(\omega \cdot \text{doy})$ | $\xi_0$ |

of $\xi$ is only reasonable if there are sufficient data available. Fischer et al. (2019) demonstrated that modeling the intra-annual variations of $\xi$ can indeed improve the GEV model. However, their model is able to combine the observations of many stations – due to additional spatial covariates – and therefore the amount of data on which the estimation is based is increased. Since in contrast we employ separate models for each station, we choose the shape parameter to remain constant $\xi = \xi_0$. For the parameters $\mu$ and $\sigma$, a variation in the form of Eq. (10) is adopted.

To be consistent, we also use a constant shape parameter in the d-GEV case. The estimation of the duration offset parameter $\theta$ is likewise associated with considerable uncertainties, because it is strongly influenced by the estimation of the parameters $\eta$ and $\sigma_0$. Eq. (5) clearly indicates this effect. Therefore, we choose for $\theta$ to remain constant $\theta = \theta_0$ as well. The parameters $\tilde{\mu}, \sigma_0$ and $\eta$ are allowed to vary periodically throughout the year according to Eq. (10).

For the maximum order of the harmonic series in Eq. (10), we choose $J = 4$. This results in a maximum of 8 regression

coefficients $\alpha_j^\varphi$ and $\beta_j^\varphi$ for each non-constant parameter. Thus, in the GEV case one would obtain one model per duration containing $3 + 2 \cdot 8 = 19$ parameters and in the d-GEV case one model describing all durations simultaneously with $5 + 3 \cdot 8 = 29$ parameters to be estimated. To reduce this number to a level where the model describes the variations sufficiently well without over-fitting, we apply a step-wise forward regression. For both the GEV model and the d-GEV model, we use the same methods to select the necessary predictor terms: As model selection criterion we use the cross-validated log-likelihood. We choose a

small number of folds $k = 2$, as recommended for cross-validation with the aim of model selection (Arlot and Celisse, 2010). In the first step of the step-wise regression, we compare all possible models that can result from the addition of the cosine term with $j = 1$ as covariate for the non-constant parameters. This yields 4 possible models for the GEV case, listed in Table 1, and analogously 8 models for the d-GEV case. The model resulting in the maximum cross-validated likelihood is retained for the next model selection step. In this next step we similarly identify the non-constant parameters for which the addition of the sine

term with $j = 1$ results in an improvement of the model. We proceed to the maximum order $J = 4$ accordingly.

For the station Bever-Talsperre, the resulting estimated d-GEV parameters are presented in Fig. 2. For comparison, the estimated parameters resulting from using a separate model for each month are shown as well. In the case of the station Bever-Talsperre, model selection yields a model with 16 parameters to be estimated. This represents a large parameter reduction compared to using one separate d-GEV model per month with $5 \cdot 12 = 60$ parameters. From Fig. 2 we can note that the choice

to keep the parameters $\xi$ and $\theta$ constant seems to be justified. In addition, the parameters $\sigma_0$ and $\eta$ show a clear variation throughout the year. The estimates of the separate models per month and the d-GEV model with covariates agree well for





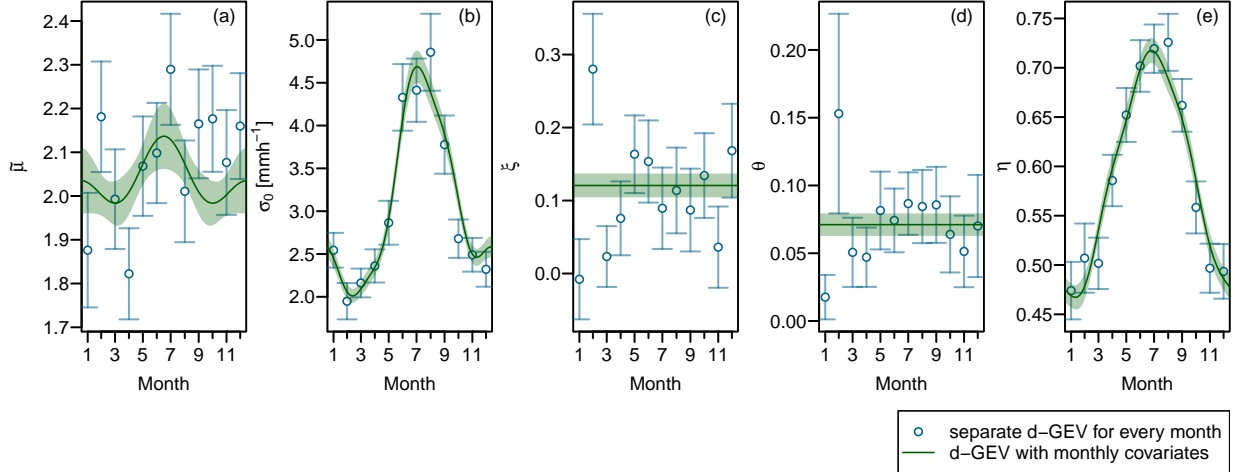

**Figure 2.** Estimated d-GEV parameters for station Bever-Talsperre: through applying one separate d-GEV model for each month (blue dots) and by modeling all month simultaneously using a d-GEV model with monthly covariates (green lines). The error bars and shaded areas show the 95% confidence intervals obtained via the estimated Fisher information matrix.

these two parameters. In the case of the modified location parameter $\tilde{\mu}$, the variations are not as pronounced as for $\sigma_0$ and $\eta$. Although it might be possible to model this parameter as constant as well, we decide not to restrict the model further in this respect.

## 2.6 Obtaining IDF curves

When modeling the annual maxima with the d-GEV distribution according to Eq. (8), we can derive IDF curves that correspond to the annual exceedance probabilities using Eq. (4). Likewise, when modeling monthly maxima using the d-GEV distribution with monthly covariates (Eq. (10)), Eq. (4) yields separate IDF curves for each month of the year. These correspond to the probabilities that a certain intensity will not be exceeded within a specific month. To distinguish between these two types of IDF curves, we refer to them as annual and monthly IDF curves, respectively.

However, we can also derive annual IDF curves, i. e. quantiles of the distribution of annual maxima, from modeling the monthly maxima. Assuming the maxima of all months in a year as independent, the non-exceedance probability $p$ of an intensity level $q_{p,d}$ within one year is derived from its monthly non-exceedance probabilities as

$$p = \prod_{m=1}^{12} G_m(q_{p,d}; \mu_m, \sigma_m, \xi_m), \tag{14}$$

for a fixed duration $d$. Therefore, to obtain the quantiles of the distribution of annual maxima, we numerically solve Eq. (14) for $q_{p,d}$. Where $q_{p,d}$ is the quantile corresponding to the exceedance probability $1 - p$, sometimes interpreted as return level associated with the return period $1/(1 - p)$. We compute $q_{p,d}$ for the entire duration range to yield the annual IDF curves.





We determine the uncertainties of the estimated IDF curves using the ordinary non-parametric bootstrap percentile method (Davison and Hinkley, 1997). We assume that considering an appropriate sampling strategy, this method accounts for the

dependence between maxima of different durations. For this purpose, all maxima from a particular year are jointly sampled in the bootstrap method, where the number of samples is set to $R = 500$. Fauer et al. (2021) demonstrated in a sampling experiment that the coverage of the 95% confidence intervals obtained in this manner stays adequate, even when the dependence of the maxima of different durations is increased.

To visualize the differences between the annual IDF curves resulting from the different models, it can be useful to compare

the parameters of the respective distributions of annual maxima. Unfortunately, these are only directly available when modeling the annual maxima. However, we can assume that the distribution of the annual maxima resulting from modeling the monthly maxima is for each duration again a GEV distribution, due to its max stability property. Thus, we estimate the GEV parameters of the distribution of annual maxima by firstly using Eq. (14) to estimate the quantiles in the range $p \in [0,1]$ and through inversion obtaining $p(q_{p,d})$. We then fit the GEV distribution to $p(q_{p,d})$ using the non-linear least squares method to estimate

$\mu_d, \sigma_d$ and $\xi_d$

$$p(q_{p,d}) \sim \exp\left\{-\left[1 + \xi_d\left(\frac{q_{p,d} - \mu_d}{\sigma_d}\right)\right]^{-1/\xi_d}\right\}. \tag{15}$$

The uncertainties of the estimates of $\mu_d, \sigma_d$ and $\xi_d$ are likewise determined using the bootstrap method.

## 2.7   Verification

We apply a verification procedure in order to asses the estimated quantiles, i.e. IDF curves. At a given station, we aim to

compare the annual IDF curves obtained by modeling the monthly maxima with those obtained by modeling the annual maxima. We model the monthly maxima using the d-GEV distribution with monthly covariates according to Eqs. (8) and (10). We abbreviate this model as monthly d-GEV in the following. For modeling the annual maxima, we use the d-GEV distribution (Eq. (8)) and abbreviate this model as annual d-GEV.

To provide a detailed analysis we follow Ulrich et al. (2020), who suggest a verification strategy that allows examining

the estimated quantiles, for each duration $d$ and probability $p$ separately. The approach is based on the comparison of the observations $o_n$ with the modeled quantile $q_p$ via the quantile score (QS) (Bentzien and Friederichs, 2014):

$$\text{QS}(p) = \frac{1}{N}\sum_{n=1}^{N} \rho_p(o_n - q_p), \quad \text{where } \rho_p(u) = \begin{cases} pu & , u \geq 0 \\ (p-1)u & , u < 0. \end{cases} \tag{16}$$

To obtain the out-of-sample performance of the model, QS is evaluated in a cross-validation setting (Wilks, 2011). For this purpose, we split the available time series at a station into $n_y$ sets, corresponding to the length of the time series in years, by

removing the maxima of all durations of a specific year $y$ for each set. The model parameters and thus the quantile $q_{p,d}$ are estimated based on the remaining data. The quantile score for a cross-validation set is calculated from $q_{p,d}$ and the respective





omitted observed annual maximum $o_{d,y}$. Therefore, the cross-validated quantile score results to:

$$\text{QS}_{\text{cv}}(p,d) = \frac{1}{n_y} \sum_{y \in Y} \rho_p(o_{d,y} - q_{p,d}). \tag{17}$$

To compare the score of the monthly d-GEV $\text{QS}_{\text{cv}}^m$ with that of the annual d-GEV $\text{QS}_{\text{cv}}^a$, we use the quantile skill index (QSI)
(Ulrich et al., 2020) that is based on the quantile skill score (Wilks, 2011):

$$\text{QSI}(p,d) = \begin{cases} 1 - \text{QS}_{\text{cv}}^m(p,d)/\text{QS}_{\text{cv}}^a(p,d) & ,\text{QS}_{\text{cv}}^m(p,d) \leq \text{QS}_{\text{cv}}^a(p,d) \\ -1 + \text{QS}_{\text{cv}}^a(p,d)/\text{QS}_{\text{cv}}^m(p,d) & ,\text{QS}_{\text{cv}}^m(p,d) > \text{QS}_{\text{cv}}^a(p,d). \end{cases} \tag{18}$$

Therefore, $\text{QSI} \in [-1,1]$, where negative values indicate a superior performance of the annual d-GEV model, whereas positive
values indicate a superior performance of the monthly d-GEV model.

## 3 Results

We first show the results for the monthly IDF curves at the station Bever-Talsperre. Based on the probability that the annual $p$-quantile is exceeded within a certain month, we investigate the seasonal variations of the IDF relationship across Germany. We
present the results in detail for six selected stations. Furthermore, we examine the annual IDF curves resulting from modeling
monthly maxima and compare them to those obtained from modeling annual maxima. We present the resulting annual IDF
curves together with the verification results for three example stations. Since the annual IDF curves derived from the monthly
maxima deviate from our original assumptions about the duration-dependence, we finally investigate the dependence of the
estimated GEV parameters on duration using annual and monthly maxima, respectively. We focus in detail on the shape
parameter.

### 3.1 Intra-annual variations

We obtain quantile estimates, i.e. IDF curves, for each month using the d-GEV distribution (Eq. (8)) with monthly covariates
(Eq. (10)), through Eq. (4). The 0.9-quantile for each month dependent on duration is shown in Fig. 3 (a) for the station Berver-Talsperre. The IDF curves exhibit a steeper slope in the summer months (pink) than in the autumn and winter months (blue).
This is related to the duration exponent $\eta$, which exhibits higher values in summer than in winter (Fig. 2 (e)). For short durations
$d \leq 1$ h the intensities reach their maximum in the summer months and their minimum in the winter months. This corresponds
to the scale offset parameter $\sigma_0$, which similarly peaks in summer (Fig. 2 (b)). In contrast, the intensity maximum of long
durations $d \geq 24$ h occurs in winter and the minimum in spring and summer, since the curves for different months intersect at
$d \approx 8$ h. The annual variation of the intensity for the different durations is presented in Fig. 3 (b). It is evident that the intensity
maximum shifts from summer for the short durations (purple/ blue) through autumn into winter for the long durations (light
green/ yellow). Here, only the quantiles of the monthly distributions for $p = 0.9$ are shown. However, the monthly quantiles
for other probability values exhibit the same behavior.

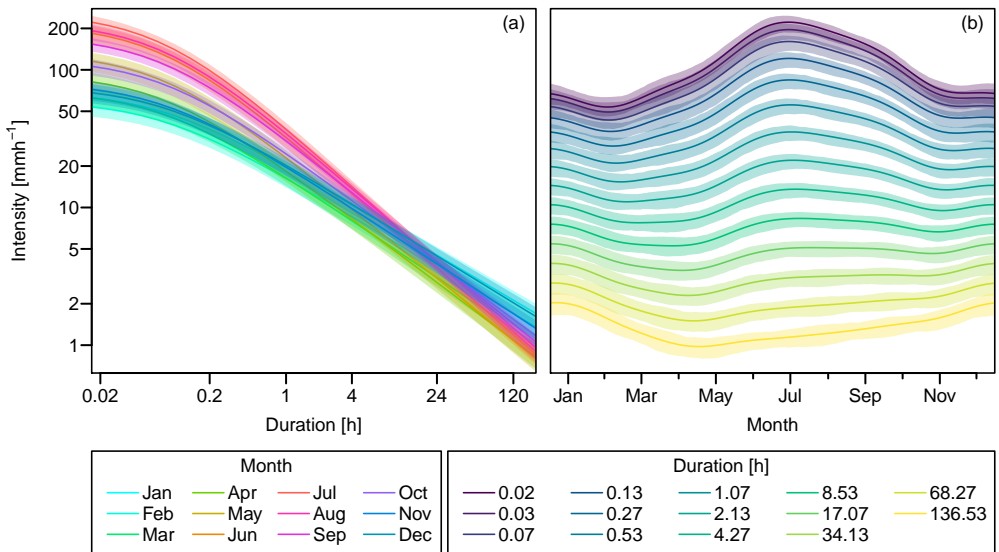

**Figure 3.** The 0.9-quantiles for station Bever-Talsperre for each month (a) and for various durations (b). The durations shown correspond to the durations $d \in 2^{\{0,1,2,\ldots,13\}}$ min discussed in Sect. 2.1. Shaded areas represent the 95% confidence intervals obtained via the bootstrap method.

Due to the exponential decrease of intensity with duration, a comparison of the $p$-quantiles of different durations is only possible on a logarithmic intensity scale. However, the interpretation of a logarithmic axis is often difficult. Therefore, in addition to the monthly 0.9-quantiles presented in Fig. 3, we will also consider the probabilities that the annual 0.9-quantile is exceeded within a given month. To do this, we first use Eq. (14) to calculate the annual quantiles $q_{p,d}$ (return values) from the monthly non-exceedance probabilities. Based on this, we calculate the probability that $q_{p,d}$ will be exceeded within a

given month. Fig. 4 (upper left) presents the probabilities that the annual 0.9-quantile $q_{0.9,d}$ (10-year return value) for different durations will be exceeded within a given month at the Bever station. This depiction is a useful complement to Fig. 3, since the probabilities for different durations vary on a linear scale, unlike the intensities. The monthly exceedance probability for short durations $d < 1\,\mathrm{h}$ (purple/ blue) exhibits a sharp peak with a maximum in July. The probability that $q_{0.9,<1\,\mathrm{h}}$ is exceeded in the months November to April is approximately zero. In the transition to longer durations, the probability decreases in July, while

a second maximum occurs in December/ January. This results for durations of about 8 h to 17 h in an extended period of time ranging from June to February, during which the probability shows similarly elevated values. For the long durations $d > 48\,\mathrm{h}$ (light green/ yellow), the probability again has one clear maximum, which occurs in December to January. The probability of $q_{0.9,>48\,\mathrm{h}}$ being exceeded in the months April to June is relatively low in this case.

    To investigate the intra-annual variations across Germany, we calculate the probability that the annual $p$-quantile is exceeded in a given month for each station. We present the results for $p = 0.9$ in Fig. 5, whereby a different choice of $p$ yields very

similar results. We summarize the information on a map by indicating the maximum probability (size of the dots), as well as the month in which the maximum occurs (color of the dots). To a certain extent, the maximum probability provides information





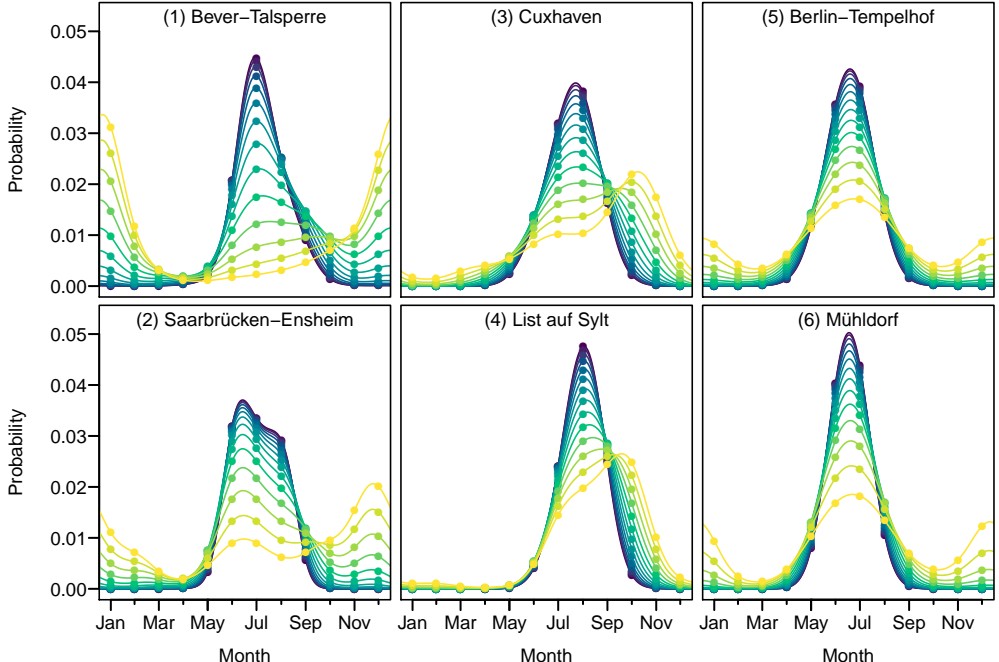

**Figure 4.** Probability of the annual 0.9-quantile $q_{0.9,d}$ being exceeded in a given month. Colors represent different durations according to the legend of Fig. 3. Dots indicate the probability that $q_{0.9,d}$ is exceeded in a certain month, while each point in a line can be interpreted as the probability that $q_{0.9,d}$ is exceeded within the surrounding block of 30 days. To illustrate the interpretation: the product of all probability values presented as dots for a chosen duration results in the annual exeedance probability $1-p=0.1$. For reasons of visual clarity, confidence intervals are not presented. Stations names are listed at the top of each plot, while the numbers indicate their positions in Fig. 5.

about the shape of the curve: A high maximum probability is associated with a narrow probability peak. This implies that the probabilities in the remaining months of the year are comparatively small. In contrast, a small maximum probability suggests

that there are other months in the year with similar probability values.

From Fig. 5 (a) it is evident that for short durations the probability peaks in summer at every station, specifically in July at most stations. There are also some stations where the maximum occurs in June or August. Noticeably, the stations where the maximum is in August are all located at the North Sea coast. Futhermore, at most stations the maximum probability is greater than 3%, which indicates a narrow probability peak in summer for short durations. When comparing the maximum probabilities

with those at $d = 24$ h (Fig. 5 (b)), we find that the maximum probability decreases considerably at almost all stations. Thus, broadening the time window within which the annual 0.9-quantile is more likely to be exceeded. At most stations, the maximum occurs in the period from June to September. At six stations, however, the maximum is reached in December or January. These stations are all located at higher altitudes (Mittelgebirge). Figure 5 (c) for $d = 120$ h reveals a comparatively heterogeneous spatial distribution, both in terms of the maximum probability and the months in which the maximum occurs. At most stations





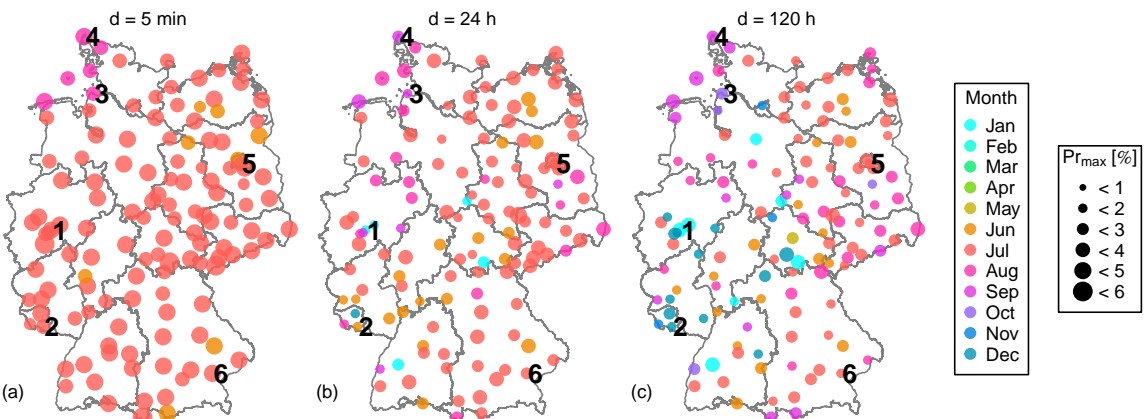

**Figure 5.** Maximum probability of the annual 0.9-quantile beeing exceeded in one month and month when the maximum occurs at each of the considered stations for three different durations. Numbers indicate the locations of example stations, presented in more detail in Fig. 4.

in Germany the maximum still lies between June and September, however, the number of stations with maximum probability between November and January is considerably increased in the west of Germany, especially in higher regions.

Regarding the month with the highest probability in the case of long durations, we derive a rough division of the stations within Germany into three types. For each type we choose two stations, for which we present the probabilities in detail in Fig. 4. We separate into stations with maximum in late autumn and winter (1,2), between September and October (3,4), and in summer

(5,6). The locations of the stations are indicated by numbers in Fig. 5. The station (1) is Bever-Talsperre, which has already been discussed in detail. Similarly to (1), the maximum at the station Saarbrücken-Ensheim (2) also shifts from summer for short durations into late autumn to winter for long durations. The stations differ, however, insofar as the maximum probability for short durations at station (2) in July varies only slightly from the probability in the months of June and September. In addition, the probability for long durations remains rather high in summer, in contrast to station (1). Stations that show a similar shift of

the maximum exceedance probability, from summer for short durations to late autumn or winter for long durations, are located exclusively in the western half of Germany and also occur mostly at higher altitudes. The exception are two stations in northern Germany. As examples for stations where the maximum occurs in September or October, for long durations, we present the monthly exceedance probabilities for Cuxhaven (3) and List auf Sylt (4). At both stations the width of the probability peak increases for long durations while its maximum shifts. This shift is more pronounced at station (3). The probabilities in the

interval between December and May are relatively low at both stations for all durations. Some stations of this type are located scattered throughout Germany. However, a clear cluster of these stations exists at the North Sea coast. Examples of stations where the probability maximum for all durations occurs in summer are Berlin-Tempelhof (5) and Mühldorf (6). An essential difference to the stations (3) and (4) is that in this case a second maximum occurs in winter for long durations. At station (6) this second maximum is even almost as high as the one in summer for $d > 120\,\text{h}$. Stations of this type occur everywhere in

Germany, but are the prominent station type in the eastern half of Germany. The example stations (5) and (6) show a very



distinct behavior for the probabilities with increasing duration. At most of the other stations of this type the signal for longer durations is less clear, as sometimes several maxima occur, or the summer maximum might be shifted by one or two months. However, the common characteristics of these stations continue to be the maximum for all durations occurring in the period between May and October and the probability for long durations showing similarly increased values throughout several months of the year.


The monthly exceedance probability is a useful indicator of the months from which the annual maxima of different durations originate. For all stations, the peak of the probability is relatively narrow for short durations, with the maximum in summer. The probability that the annual 0.9-quantile occurs in one of the other seasons is negligible. This fact contradicts the assumption of the block maxima approach that precipitation intensities are identically distributed within the block of one year. In other words, a block size of one year for short durations results in a much smaller effective block size of about four to six months. With respect to longer durations, the stations differ greatly, but it can be generally stated that the effective block size increases for long durations. Thus, the annual maxima at a station for different durations originate from effective blocks of different sizes, which might even be in different seasons, depending on the stations location. This effect is further emphasized in Fig. B1. Modeling the monthly maxima, on the other hand, avoids this problem. Therefore, in the following chapter we compare the annual IDF curves derived from annual maxima with those derived from monthly maxima.


### 3.2 Annual IDF curves

We obtain the annual IDF curves and their confidence intervals from modeling annual and monthly maxima, using the respective methods described in Sect. 2.6. We compare the estimated quantiles from both models along with the QSI described in Sect. 2.7. Figure 6 presents the IDF curves (lower panels) together with the QSI (upper panels) for three example stations. In addition to the station Bever-Talsperre (1), with the longest time series, we present the results for the stations Saarbrücken-Ensheim (2) and Cuxhaven (3), since these three stations cover a broad spectrum in terms of differences between the quantile estimates obtained by both models as well as their uncertainties. The QSI is used to compare the performance of both models. Positive values (red) indicate an increase in the skill of the monthly d-GEV model compared to the annual d-GEV model, while negative values (blue) indicate that the annual model is superior. The result of the QSI may be less reliable if the length of the time series $T$ is shorter than the period corresponding to the non-exeedance probability $p = 1 - 1/T$ being verified. Therefore, we indicate the length of the time series available for verification by dots in the upper panels in Fig. 6.

For the station Bever-Talsperre, the IDF curves resulting from the two different models are almost identical over a long duration range $d < 8\,\mathrm{h}$. Therefore, in this duration range the models performances differ only marginally, indicated by $|\mathrm{QSI}| \leq 0.05$ for almost all probabilities. For high probabilities $p \geq 0.98$, the QSI suggests a slightly better performance of the monthly model for durations $4\,\mathrm{min} \leq d \lesssim 2\,\mathrm{h}$. At $d \approx 8\,\mathrm{h}$ the IDF curves of the monthly model start to deviate from those of the annual model. More precisely, the IDF curves of the monthly model no longer exhibit a power-law behavior for $d > 8\,\mathrm{h}$ but decrease more gradually. Due to the larger differences in the IDF curves, the models performances vary considerably ($|\mathrm{QSI}| > 0.05$) in this range. However, the sign of the QSI differs for $d \lesssim 34\,\mathrm{h}$ and $d \gtrsim 68\,\mathrm{h}$: For $d \lesssim 34\,\mathrm{h}$, the data are better represented by the annual model, as indicated by the negative values of the QSI in this range. For $d \gtrsim 68\,\mathrm{h}$ and $p \geq 0.95$, however, the monthly





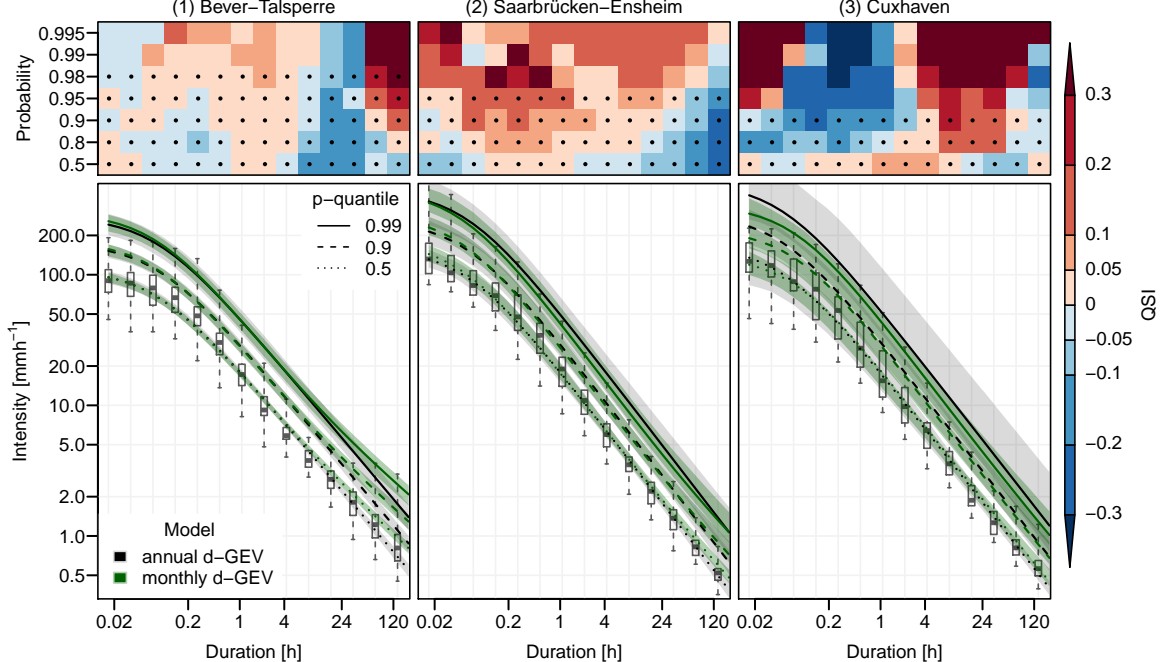

**Figure 6.** Annual IDF curves for three example stations estimated via two models: modeling annual maxima with the d-GEV distribution and modeling monthly maxima using the d-GEV distribution with monthly covariates. The shaded areas represent the respective 95% confidence intervals. The distributions of the observed annual maxima are shown as box-whisker plots, where the whiskers cover the complete data range. In the upper panels the corresponding OSI values are presented, indicating the comparison of the models' performances.

model is a considerable improvement over the annual model. This is evident both from the strongly positive QSI values in this range as well as directly from the data, shown as box-whisker-plots, as the maximum of the observations extends above the modeled 0.99-quantile. The models do not differ much with respect to the width of the 95% confidence intervals.

At station Saarbrücken-Ensheim, the differences in the IDF curves of the two models are more pronounced throughout the entire duration range: the estimated quantiles of the monthly model are higher for very small and very large durations, but

lower in the range $8\,\mathrm{min} \lesssim d \lesssim 48\,\mathrm{h}$ than those of the annual model. Thus, again the monthly model does not comply with a power law. The QSI indicates that the monthly model is mostly an improvement, except for smaller probabilities at longer durations. Since this station provides a shorter time series than station Bever-Talsperre, i.e. only 24 years, the 95% confidence intervals of the annual model are wider, especially for short durations and high probabilities. This indicates that the monthly model benefits from utilizing more data regarding the uncertainties.

This appears even more prominent at the Cuxhaven station. Since the high-resolution time series at this station covers only 19 years, the uncertainties of the annual model are considerably wider than those of the monthly model. The estimates of the monthly model for the 0.9- quantile and the 0.99-quantile are below the respective estimates of the annual model. The estimates for the 0.5-quantile differ only slightly. The quantiles of the monthly model roughly parallel those of the annual





model for longer durations. Thus, the monthly model does not deviate essentially from a power-law at this station. The QSI
does not provide a clear indication regarding which model better represents the data, but fluctuates between positive and
negative values. This seems to be in agreement with the observations. The spread of the boxes and whiskers first increases and
then decreases over duration. As a result, in the duration ranges with narrower box-whisker plots, the monthly model better
represents the data, especially for higher quantiles, while the annual model is more suitable particularly for $8\,\mathrm{min} \leq d \lesssim 1\,\mathrm{h}$,
where the boxes and whiskers are rather broad.

Overall, we find that the differences between the annual and the monthly model are very heterogeneous for individual
stations. In terms of model performance, it is also not possible to make a general statement as to when the monthly model
might be superior to the annual model. Although the three presented stations already illustrate a broad spectrum of possible
differences between the models, they cannot be considered a complete representation of the very diverse results at the 132
stations in Germany regarded in this study. For example, modeling monthly maxima may also result in higher quantile estimates
over the entire duration range than modeling annual maxima. Additionally, a number of stations occur for which modeling the
monthly maxima leads to absolutely no gain in model skill. However, two general statements can be made:

1. Modeling monthly maxima provides a clear improvement in terms of the quantile estimates' uncertainties, especially for
stations with short observational time series.

2. Although a power-law behavior for long durations is assumed for the monthly IDF relations, the resulting annual IDF
curves can deviate from this behavior and are therefore more flexible.

Since we aim to better understand this deviation from the original assumptions in Eqs. (5)-(7), in the following section we
examine the relationships between the GEV parameters and duration that follow from modeling monthly maxima.

### 3.3 Duration dependence

To model the IDF relationship, we have so far assumed that the GEV parameters depend on duration according to Eqs. (5)-(7).
This results in a power law behavior, or so-called simple scaling, of intensity with duration except for short durations $d < 1\,\mathrm{h}$.
The curvature of the IDF curves for short durations (in a double-logarithmic plot) is controlled by the parameter $\theta$. Fig. 3 (a)
illustrates that the IDF curves for each month follow this imposed pattern. However, the resulting annual IDF curves, shown
in Fig. 6 in green, deviate from this behavior and thus from the assumptions in Eqs. (5)-(7). To investigate this deviation in
more detail, we estimate the annual GEV parameters resulting from modeling monthly maxima, using nonlinear regression
according to Eq. (15) separately for each duration. We use the obtained parameters only to compare them with those estimated
directly from modeling the annual maxima. They are not intended as a basis of any further analysis. We compare the following
four models:

    – modeling annual maxima using

        – a separate GEV distribution for each duration (annual GEV)

420         – one d-GEV distribution (annual d-GEV)





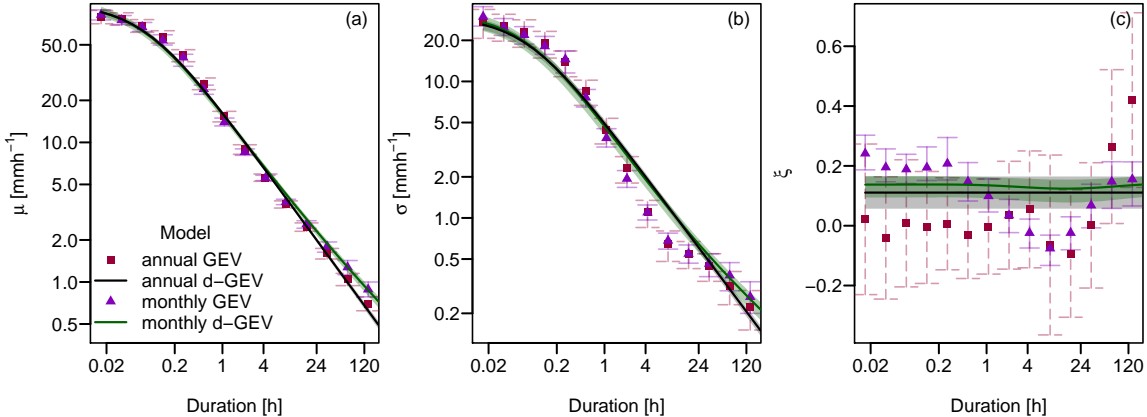

**Figure 7.** Annual GEV parameters $\mu, \sigma, \xi$ for station Bever-Talsperre estimated via four different models along with 95% confidence intervals shown as error bars and shaded areas.

- modeling monthly maxima using

    - a separate GEV distribution with monthly covariates for each duration (monthly GEV)

    - one d-GEV distribution with monthly covariates (monthly d-GEV).

The annual GEV parameters $\mu, \sigma, \xi$ estimated via these models are presented depending on duration for the station Bever-

Talsperre in Fig. 7 including their respective 95% confidence intervals. The uncertainties of the parameters estimated directly from the annual maxima can be derived using the Fisher information matrix estimated during optimization, see Sect. 2.4, while the uncertainties of the parameters estimated by modeling the monthly maxima are determined using the bootstrap method, as described in Sect. 2.6.

For the location parameter $\mu$ (Fig. 7 (a)), the estimates of both annual models (red squares and black line) agree well, i.e.

$\mu$ follows a power-law for durations $d \geq 1\,\mathrm{h}$, while the curve decreases more gradually for shorter durations. The estimates of the monthly models (purple triangles and green line) are consistent with those of the annual models for durations $d \lesssim 24\,\mathrm{h}$, however, both monthly models agree on a slower decline of $\mu$ for longer durations $d \gtrsim 24\,\mathrm{h}$ and thus a deviation from simple scaling. We observe a quite similar behavior of the model estimates for the scale parameter $\sigma$ (Fig. 7 (b)): The estimates of all models agree relatively well for short durations $d \lesssim 1\,\mathrm{h}$ and both monthly models show an upwards deviation from simple

scaling for longer durations $d \gtrsim 24\,\mathrm{h}$. However, the estimates of both GEV models (squares and triangles) deviate noticeably from simple scaling towards smaller values in the range $1\,\mathrm{h} \lesssim d \lesssim 24\,\mathrm{h}$. Regarding the uncertainties for the estimates of $\mu$ and $\sigma$, the annual GEV model (red) is associated with the largest uncertainties. By considering more data, the parameters can be estimated more accurately using the monthly GEV model (purple). Similarly, the annual d-GEV model (black) exhibits considerably smaller uncertainties than the annual GEV model because here the addition of data from other aggregation levels

leads to a more confident estimate. Yet, the estimates of the monthly d-GEV model (green) present greater uncertainties than those of the annual d-GEV model, even though this model allows using the most data, namely monthly maxima of all durations.





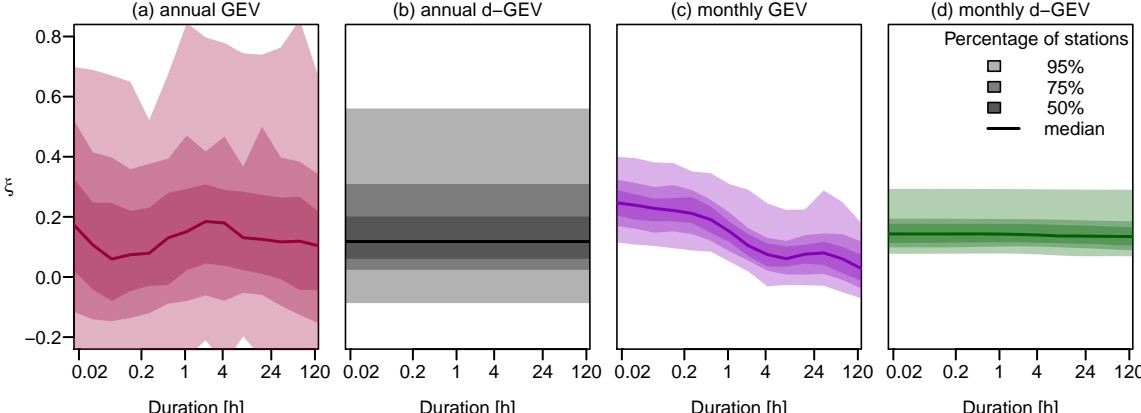

**Figure 8.** Distribution of shape parameter $\xi$ estimated using four different models (columns) at all 132 stations. The shaded areas present the percentage of stations for which the estimated value lies within the respective range while the line depicts the median value.

We suspect that this is due to the two consecutive modeling steps used to estimate the parameters. The estimates for $\mu$ and $\sigma$ obtained from all four models agree relatively well at most of the other stations (not presented). Similar to the station Bever-Talsperre, at several stations the estimates of the monthly models show an upward deviation from simple scaling for long durations. The estimates are associated with considerably larger uncertainties for most other stations, where shorter time series are available.

The shape parameter $\xi$ determines the tail of the distribution and its estimation is therefore subject to relatively large uncertainties depending on the length of the time series. This is similarly the case for the station Bever-Talsperre (Fig. 7 (c)) with a time series of 51 years, where the estimates of $\xi$ based on annual maxima of a single duration (red squares) vary in the range 0.09 to 0.4, but since the 95% confidence intervals are very broad, it is challenging to derive a relationship between $\xi$ and duration. For the d-GEV we therefore assume a constant shape parameter (Eq. (7)), which for the annual d-GEV model (black line) is estimated to be 0.11. Compared to the separate estimates of the annual GEV for each duration (red squares), this value appears reasonable. Additionally, the uncertainties decrease significantly when modeling the annual maxima of all durations simultaneously. For the monthly d-GEV, $\xi$ is also assumed to remain constant over duration for each month. In addition, we hypothesize that $\xi$ does not change throughout the year either, as described in Sect. 2.5. Interestingly, $\xi(d)$ resulting from the monthly d-GEV model (green line) nevertheless deviates slightly from being constant. The estimated values are slightly higher than that of the annual d-GEV model and vary between 0.12 and 0.14, with the minimum at about $d \approx 17\,\mathrm{h}$. Contrary to $\mu$ and $\sigma$, the uncertainties of the monthly d-GEV model (green) for the estimation of $\xi$ are smaller than those of the annual d-GEV model (black). When we model the monthly maxima for each duration separately, we benefit from being able to estimate the shape parameter with smaller uncertainties without constraining the relation between $\xi$ and duration. Thus, we observe a distinct variation of $\xi$ estimated via the monthly GEV model (purple triangles) over duration. The estimated values exhibit a minimum, similar to those of the monthly d-GEV model (green line), at $d \approx 8\,\mathrm{h}$ and vary in the range -0.08 to 0.24.





Since the shape parameter estimates of the four different models vary substantially at the individual stations, we summarize the information for all stations in Fig. 8. For each model, the distribution of the estimated value over all stations is plotted with
respect to duration. We observe that $\xi$ estimated using the annual GEV model (red) varies widely among stations. The model seems to only provide estimates in a reasonable range at 75% of the stations based on the small amount of data. The median of all stations appears to oscillate around a constant value. For the annual GEV model (black), the range in which the estimates vary among stations is narrower, but implausibly high values still occur at some stations. The median is consistent with the median of the annual GEV model (red). In comparison, the range in which the estimated values of the monthly d-GEV model
(green) vary is much smaller and the values are within a reasonable range at all stations. Although in this model $\xi$ is assumed to be constant over duration and months, we see a subtle decrease in the median with duration. Likewise, the variation for $\xi$ between individual stations is smaller for the monthly GEV model (purple) than for either of the annual models (red and black). Additionally, we can see a clear decrease of the shape parameter with duration for this model, since in this case no relation between $\xi$ and duration is predefined.

## 4   Discussion

We model the monthly maxima of precipitation intensity for a range of durations at 132 stations. This allows us to investigate seasonal variations in the IDF curves throughout Germany. We find that the IDF curves are steeper in the summer months than in the winter months, corresponding to a larger value of the duration exponent $\eta$ in summer compared to winter, see Fig. 2 (e). As an example, we present the estimated 0.9-quantiles for each month at the station Bever-Talsperre with the longest observation
time series in Fig. 3; the results are similar for other probability values and all other stations. Willems (2000) determined the rate of exponential decline $\beta$ of intensity with duration (equivalent to $\eta$) for different seasons and likewise found that $\beta$ exhibits the smallest values in the winter season. At the station Bever-Talsperre, the different duration exponents $\eta$ cause the IDF curves of the different months to intersect at $d \approx 8\,\mathrm{h}$, resulting in a shift of intensity maximum from summer for short durations to winter for long durations, see Fig. 3. This intersection of the IDF curves and the resulting change in seasonality for different durations
is not generally present at all stations in the investigated duration range. However, since different duration exponents of the curves in summer and winter occur at all stations, we suspect that at all stations, given sufficiently long durations, the intensity maximum moves into winter eventually. The change in seasonality for different durations can be illustrated particularly well when considering the probability of the annual $p$-quantile $q_{p,d}$ being exceeded within a given month. We present the results for the monthly exceedance probability of the annual 0.9-quantile, i.e. 10-year return level, in Figs. 4 and 5, since sufficient
data are available at each station, however, we find that other values for $p$ yield similar results. We see that the probability has a narrow peak at all stations in Germany with a maximum between June and August. This means that extreme events of short duration, i.e. caused by convective precipitation cells, are likely to occur in summer while the probability of these events occurring in the months October to April is very small. This is consistent with the results for one station in Belgium (Willems, 2000). In the transition to longer durations, the probability decreases in the summer months and increases in the remaining
months. Therefore, extreme events with a duration of several hours up to about one day are more likely to occur within a longer

 

time period or even spread throughout the whole year, depending on the station. Finally, there are large differences within Germany with respect to the months in which long-lasting extreme events, i.e. frontal events, are more likely to occur.

We roughly divide the stations into three groups: stations where extreme events with a duration of 1.5 days or longer occur (1) most likely in late fall or winter, (2) most likely from summer to fall and (3) spread throughout the year. Exemplary for the first group we present the stations Bever-Talsperre and Saarbrücken-Ensheim in the left column of Fig. 4. Stations of this group are located exclusively in the western half of Germany and mainly in higher altitude areas. The location of these stations coincides well with the results of Fischer et al. (2018) with the exception of two stations in northern Germany. They explain the increased occurrence of longer-lasting extreme events in these regions in late autumn and winter with the stronger westerly winds during these months, which cause particularly high precipitation amounts on the windward sides of the mountain chains (Mittelgebirge) due to the forced uplift of the air. However, this study is based on daily precipitation sums. The fact that when modeling several durations simultaneously, the seasonal variations are observed at longer durations than when modeling a single duration might be due to the smoothing of the seasonal signal. A distinct cluster of stations from the second group is located along the North Sea coast. As examples, we present Cuxhaven and List auf Sylt in the middle column of Fig. 4. This group is also characterized by extreme convective precipitation events occurring most likely in August, which could be related to the water temperature in this region reaching its maximum during this month. Accordingly, a possible explanation for the high probability of long-lasting heavy precipitation in the following months might be that extra-tropical cyclones transport air, which was warmed over the North Sea and thus features a high water content, into this region. Finally, we sort most stations in Germany into group three. As examples we present Berlin-Tempelhof and Mühldorf in Fig. 4 on the right. The stations in this group share the following common characteristic: Extreme events of all durations are most likely to occur between May and October. However, while short events are considerably more likely to occur solely in summer, long events may occur over longer periods of time or even spread throughout the year.

Consequently, as a next step we investigate the extent to which the annual IDF curves are affected by modeling the monthly maxima, since the effective block size from which the annual maxima originate changes for different durations. For this purpose, we compare modeling the monthly maxima with modeling the annual maxima in terms of the resulting quantile estimates as well as the model performance. Exemplary, we present the results for three stations in Fig. 6. We find major differences between individual stations, however in general modeling the monthly maxima presents a substantial improvement in terms of uncertainties of the quantile estimates, especially for stations that provide only short observational time series. Furthermore, it is noticeable that although a power-law behavior is assumed for the monthly IDF relationships, the resulting annual IDF curves may deviate from this behavior, especially for long durations $d \gtrsim 24$h. This result is consistent with the findings of Willems (2000) who observes a decrease in the rate of exponential decline of intensity with increasing duration. We observe this decrease particularly pronounced at the station Bever-Talsperre, where we also find a clear shift of the seasons in which extreme events of different durations occur. We therefore suspect that the lower slope of the annual IDF curves at long durations is related to this shift.

To compare the performance of both models, we calculate the quantile score (QS) respectively in a cross-validation setting, see Sect. 2.7. However, the results for high quantiles may be less reliable, depending on the length of the time series available



for verification. Nevertheless, this is a general problem in the verification of very rare events and is thus independent of the verification score. We observe good agreement in the resulting values of the quantile skill index (QSI) with the visual comparison of the modeled quantiles with the empirical quantiles of the observations. Therefore, we consider the QSI to be an appropriate measure to analyze the model performance in detail, i.e. for individual probabilities and durations. We cannot draw

general conclusions concerning the model performance. At some stations, such as Cuxhaven, we find that modeling monthly maxima improves the estimates of the annual IDF curves for almost all probabilities and durations. However, at many stations the improvement in the estimation is limited to a selected range of probabilities and durations, and there also exist stations at which the estimated quantiles of the monthly model are always worse than those of the annual model. Since the objective of this study is to model the seasonal variations at all stations by applying a uniform framework, the model selection was

performed identically for all stations, e.g. choosing $\theta = $ const. and $\xi = $ const.. This results in varying quality of representation of the parameters at different stations. We expect that parameter estimation for rare events should generally improve, as the introduction of smooth variation during the year allows for the inclusion of additional information. Therefore we assume that with more focus on estimating the IDF curves of a single station, and thus a more targeted choice of model selection approach and initial conditions, the model performance of the monthly model at this station can be considerably improved.

Instead of using the more complex modeling of monthly maxima to estimate annual IDF curves, one might also try to implement the resulting characteristics directly into the model for annual maxima. We therefore investigate what conclusions we can draw from the modeling of monthly maxima about the behavior of the parameters of the distribution of annual maxima depending on duration. Thereby, we find that for some stations the location $\mu$ and scale $\sigma$ parameters deviate from the assumption of simple scaling toward higher values for long durations. Fauer et al. (2021) showed that this behavior of the parameters

can be modeled by an additional parameter $\tau$, called intensity offset. They report that the addition of this parameter for the stations of the Wupper-Catchment, in which the example station Bever-Talsperre is located, leads on average to an improved estimation of the annual IDF curves for medium to long durations.

    With respect to the shape parameter $\xi$, we find that modeling monthly maxima leads to much smaller uncertainties in the estimation of this parameter due to the larger amount of available data. Analogously, we also observe that $\xi$ estimates based on

the monthly maxima vary in a smaller, and indeed more reasonable, range at all stations. When modeling the monthly maxima of each duration separately, we do not specify an explicit dependency of the GEV parameters on duration and therefore we can identify how $\xi$ behaves as a function of duration. At station Bever-Talsperre, we observe a pronounced variation of the resulting $\xi$ estimates over duration, with a minimum at about $d \approx 8\,h$. To evaluate this result, we can visually examine the distribution of annual maxima for different durations, seen in Fig. B1 and in Fig. 6 (bottom left) as box-whisker plots. It is noticeable that the

observations for $d \approx 8\,h$ actually cover the smallest range, thus agreeing with the results for the shape parameter. On average over all stations, we find that the shape parameter decreases with duration, reaching values around zero for most stations at long durations. We could imagine that the explicit modeling of this decrease of $\xi$ yields similar results to assuming a different duration exponent for the parameters $\mu$ and $\sigma$ of the d-GEV, so called multi-scaling (Gupta and Waymire, 1990; Van de Vyver, 2018). Possibly the latter implementation could be beneficial, since the estimation of these parameters is associated with less

uncertainty than that of $\xi$.





## 5   Conclusions

This study focuses on modeling the intra-annual variations of extreme precipitation on different time scales. For this purpose, we employ a duration-dependent generalized extreme value (d-GEV) distribution with monthly covariates. Using this approach, we can investigate seasonal variations in the intensity–duration–frequency (IDF) relationship while also obtaining more reliable 570   estimates for the annual IDF curves. This is accomplished by utilizing information on extreme events more efficiently. We find that everywhere in Germany the short convective extreme events are most likely to occur in the summer months, whereas there are regional differences for the seasonality of long-lasting stratiform extreme events. Our findings will allow future studies to identify meaningful factors accounting for these regional differences. Furthermore, our results show that the annual IDF curves based on the monthly maxima constitute a major improvement in terms of uncertainties of the estimates. Using the 575   quantile skill index (QSI), we compare the performance of the models based on the annual and monthly maxima and show that, for some stations, modeling the monthly maxima also leads to a considerable improvement in this regard. A limitation of this study are the strict assumptions that were imposed on the seasonal variations of the distribution parameters. Subsequent studies should therefore investigate the degree to which relaxing these assumptions might further improve the performance of the model based on monthly data. For example, in the framework of a vector generalized additive model (Yee and Stephenson, 580   2007) it would be possible to model these smooth variations in a non-parametric form. Finally, we can demonstrate that at some stations the annual IDF curves based on the monthly maxima deviate from the assumption of scale invariance for long durations. We illustrate that this behavior can be captured by a different parameterization of the location and scale parameter. For future research, it might be of interest to compare the monthly model employed in this study with an annual model that uses different parameterization, e.g. the one proposed by Fauer et al. (2021). Moreover, by including additional information 585   in the form of smooth variations during the year, we observe that the shape parameter decreases with duration when averaged over all stations. Based on this result, future research should investigate whether the assumption of a constant shape parameter is appropriate for a wide range of durations from minutes to several days, or whether a more appropriate explicit relationship can be identified.

In conclusion, the use of monthly maxima can be beneficial in several respects when estimating IDF curves, even when 590   information on seasonal variations is not required.

*Code and data availability.* The station data are mostly publicly available via the climate data center of the DWD (ftp://ftp-cdc.dwd.de/ climate_environment/CDC/observations_germany/climate). We provide the monthly maxima for 14 aggregation times at all 132 stations, serving as the basis for the analysis, online (Ulrich et al., 2021). The statistical analysis was performed using the package `IDF` for the `R` environment (Ulrich et al., 2020; R Core Team, 2020). The package is available for download at https://cran.r-project.org/web/packages/IDF.



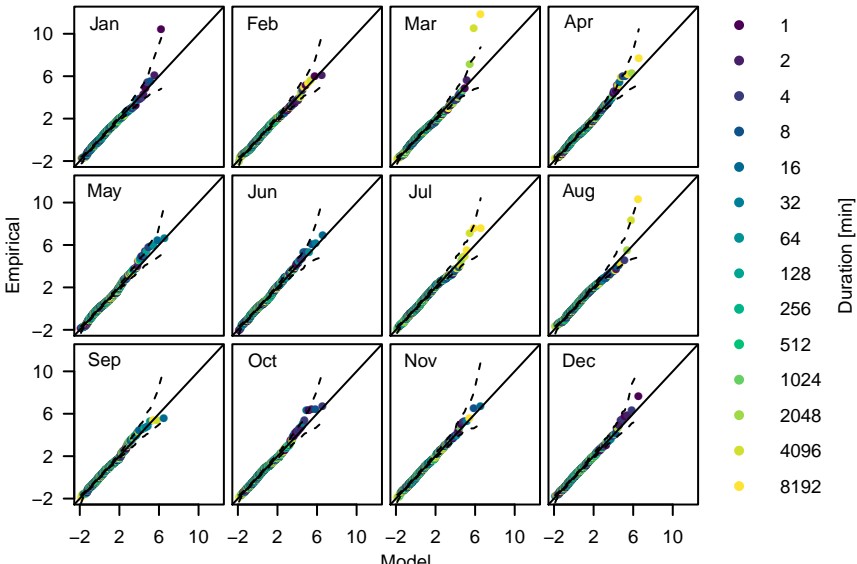

**Figure A1.** Diagnostic q-q plots of the d-GEV model for each month at station Bever-Talsperre. The observations and the modeled quantiles are transformed to standard Gumbel $G(\mu = 0, \sigma = 1, \xi = 0)$ to remove the duration dependency. Dashed lines represent 95% confidence intervals.

## Appendix A: Model diagnosis for station Bever-Talsperre

The distribution of monthly maxima for a range of durations is modeled using the d-GEV distribution. We visually inspect, whether the d-GEV distribution is a reasonable approximation for the distribution of monthly maxima at the considered stations using quantile-quantile (q-q) plots. As an example, we present the q-q plots for the station Bever-Talsperre with respect to each month in Fig. A1. The different aggregation times are indicated by different colors. We find that the d-GEV distribution describes the monthly maxima sufficiently well for each month. There are few outliers that correspond to the limits of the duration range, i.e. very short or very long durations.

## Appendix B: Annual maxima station Bever-Talsperre

The annual maxima of different durations for the station Bever-Talsperre are presented in Fig. B1. The colors indicate the month in which the respective maxima occurred. It is evident that at this station the maxima of short durations occur mostly in summer, while from $d \gtrsim 8\,\mathrm{h}$ the maxima originate from different seasons, especially autumn and winter. This is in agreement with the results presented in Fig. 4 (top left). In addition, we find that the span of the data exhibits a minimum at $d \approx 8\,\mathrm{h}$. This is consistent with the minimum value of the shape parameter $\xi$ at this duration observed in Fig. 7 (right).



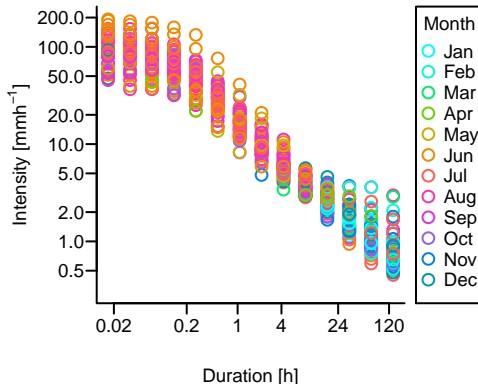

**Figure B1.** Annual intensity maxima for different durations at station Bever-Talsperre. The months in which the respective maxima occurred are represented by colors.

*Author contributions.* H.R. and J.U. conceptualized this study. H.R. provided supervision. J.U. developed the software, carried out the statistical analysis and evaluated the results. F.F. supported the code implementation. H.R. and F.F. contributed to the interpretation of the results. J.U. prepared the original draft. H.R. and F.F. revised the draft.

610

*Competing interests.* The authors declare that they have no conflict of interest.

*Acknowledgements.* This study was developed within the framework of the research training program NatRiskChange funded by the Deutsche Forschungsgemeinschaft (DFG; GRK2043/1 and GRK2043/2) at Potsdam University. Is was partially supported through grant CRC 1114 "Scaling Cascades in Complex Systems", Project A01 "Coupling a multiscale stochastic precipitation model to large scale flow" as well as

615

the *ClimXtreme* project (Grant number 01LP1902H).

The authors would like to thank the Wupperverband as well as the Climate Data Center of the DWD, for providing and maintaining the precipitation time series. J.U. kindly appreciates the support and motivation offered by Lisa Berghäuser and Oscar E. Jurado.



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
