# Peer review of "Modeling seasonal variations of extreme rainfall on different time scales in Germany"

_Hydrology and Earth System Sciences, 2021_

## Author Comment (AC1)

**Reply on RC1**

Jana Ulrich, Felix S. Fauer and Henning W. Rust

August 20, 2021

Dear Reviewer,

Thank you very much for your informative and helpful comments. We address them point by point below.

**General comments**

The general point concerns the small block size of one month for the extreme value analyses. Even if the GEV is fitting well as shown by the authors the basic assumption of sufficient large n for the validity of Fisher–Tippett–Gnedenko theorem might be questionable also considering the fact that in certain months of some years no extremes might be observed. This problem becomes especially relevant given the results from which months the maxima from different durations originate with respect to the effective block size. Especially the latter is much larger than one month for long durations. I would suggest to discuss this a little further, may be also considering future research regarding specific analyses periods/ seasonal sub-divisions depending on the duration.

**Answer**

You are correct that a detailed analysis of the monthly block size is needed, however this has been addressed in previous studies (Maraun et al., 2009; Rust et al., 2009; Fischer et al., 2018). The choice of block size typically involves finding a balance between rapid convergence to the GEV distribution and a sufficiently large number of block maxima for parameter estimation.
Since the aim of our study is to resolve the seasonal variations, we need to choose a sub-annual block size. We used q-q plots to investigate whether the monthly maxima can be well described by the d-GEV model. We present the q-q plots for the example station Bever-Talsperre again below in Figure 1. The plots show that the d-GEV distribution describes the monthly maxima of the different durations sufficiently well. To investigate the influence of the choice of a larger block size, we additionally provide the q-q plots for a block size of 2 and 3 months, respectively, in Figure 2. Figures 1 and 2 illustrate that with increasing block size information about the seasonal variations is reduced, while an increasing block size does not lead to a significant improvement of the model fit.
We agree with your concern that extreme values may not occur in certain months of the year, depending on the region under investigation. Indeed, it could be problematic to consider monthly maxima if there is no precipitation occurring at all during certain months. Nevertheless, this is not the case in Germany. You are right that Figure 4 in the manuscript indicates that for certain months the exceedance probabilities of the annual maximum are very low. But this should not be interpreted as a sign that in these months no extreme precipitation values occur. Rather, it means that the precipitation intensities, which are extreme in these months, are comparatively low with respect to the whole year. To illustrate that the GEV distribution describes the monthly maxima of individual durations sufficiently well and that a larger block size does not lead to a significant improvement in the model fit even when modeling individual durations, we additionally provide the q-q plots for two selected durations in Figures 3-6. Consideration of the remaining durations led to the same results.

[Figure]

Figure 1: Diagnostic q-q plots of the d-GEV model for each month at station Bever-Talsperre as provided in Fig. A1 of the manuscript . The observations and the modeled quantiles are transformed to standard Gumbel $G(\mu = 0, \sigma = 1, \xi = 0)$ to remove the duration dependency. Dashed lines represent 95% confidence intervals.

[Figure]

Figure 2: Diagnostic q-q plots similar to Figure 1, but for a block size of 2 months (left) and 3 months (right).

[Figure]

Figure 3: Diagnostic q-q plots of the GEV model for a duration of 1 minute for each month at station Bever-Talsperre. Again, the observations and the modeled quantiles are transformed to standard Gumbel $G(\mu = 0, \sigma = 1, \xi = 0)$.

[Figure]

Figure 4: Diagnostic q-q plots similar to Figure 3, but for a block size of 2 months (left) and 3 months (right).

[Figure]

Figure 5: Diagnostic q-q plots of the GEV model for a duration of 2048 minutes ($\approx$ 34 hours) for each month at station Bever-Talsperre.

[Figure]

Figure 6: Diagnostic q-q plots similar to Figure 5, but for a block size of 2 months (left) and 3 months (right).

**References**

- Fischer, M., Rust, H. W., and Ulbrich, U.: Seasonal Cycle in German Daily Precipitation Extremes, Meteorol. Z., 27, 3–13, https://doi.org/10.1127/metz/2017/0845, 2018.

- Maraun, D., Rust, H. W., and Osborn, T. J.: The annual cycle of heavy precipitation across the United Kingdom: a model based on extreme value statistics, Int. J. of Climatol., 29, 1731–1744, https://doi.org/10.1002/joc.1811, 2009.

- Rust, H., Maraun, D., and Osborn, T.: Modelling seasonality in extreme precipitation, Eur. Phys. J. Spec. Top., 174, 99–111, https://doi.org/10.1140/epjst/e2009-01093-7, 2009.

**Specific comments**

**1.**

Please indicate the specific GEV parameters on the figures of relate a) – e) to the specific parameters in the caption.

**Answer:** In Figure 2 in the manuscript, the d-GEV parameters are indicated at the y-axis of each of the plots (a)-(e), but we can also add this informtaion to the caption.

**2.**

Lines 233ff: Please add some more explanation how the bootstrap is carried out.

**Answer:** Thank you for pointing out that the bootstrapping procedure is not sufficiently explained. We will improve the explanation.

**3.**

Equation 16: The equation is not completely clear to me. The variable u is not explicitly defined. However, if I assume $u = (o_n - q_p)$ then it should read e.g. $\rho_p = pu$ and not $\rho_p(u) = pu$?

**Answer:** You are right, the explanation of the Quantile Score is very short. We will add the information that $\rho_p(u)$ is the check loss function where for the Quantile Score the arguments is $u = o_n - q_p$.

**4.**

Lines 425ff: The uncertainties are estimated with different methods, the Fisher information matrix and the bootstrap method. Are these results comparable? Why not using bootstrap for all uncertainty assessments?

**Answer:** Thank you for pointing this out. You are correct that it is better for reasons of comparability to estimate all uncertainties using the same method. We will change this and estimate all uncertainties via the boostrap method in Figure 7.

---

## Author Comment (AC2)

**Reply on RC2**

Jana Ulrich, Felix S. Fauer and Henning W. Rust

August 24, 2021

Dear Reviewer,

Thank you very much for your detailed and helpful comments. We address them point by point below.

**Main comments**

**1.**

The article is quite long and there are mainly repetitions. In particular the "Discussion" section is very mainly an extended summary of the results. There are actually few sentences of discussion. Would it be possible to drastically shorten this part?

**Answer:** You are right, there are many repetitions in the discussion. We believe they could be avoided by merging the sections results and discussion (as suggested in RC3) and we will change the manuscript accordingly.

**2.** I'm a bit puzzled by the fact that a constant xi is considered in the d-GEV distribution. As Fig. 8 shows, the xi parameter seems to be decreasing with duration (as expected). The authors claim that the shape parameter in difficult to estimate, which is true, but don't you think that a simple model on xi (e.g. log-linear wrt duration) could be managable?

**Answer:** Thank you for this interesting remark. For the dependence of the GEV parameters on duration in equations (5)-(7), we followed the assumptions of Koutsoyiannis et al. (1998). These assumptions are derived on the basis of an empirical formulation for the relationship between precipitation intensity, duration, and frequency, which is not based on any theoretical reasoning. Based on annual maxima, it is difficult to find an alternative formulation for the dependence of the shape parameter on duration $\xi(d)$, since the estimation of this parameter is associated with large uncertainties. This is also evident in our manuscript from Figures 7(c) (red squares) and 8(a). To the best of our knowledge, previous studies on estimating IDF curves have therefore always used a constant shape parameter $\xi(d)$ =const. within a duration-dependent extreme value distribution (Koutsoyiannis et al., 1998; Van de Vyver and Demarée, 2010; Lehmann et al., 2013; Van de Vyver, 2015; Van de Vyver, 2018). One important outcome of our study is that when monthly maxima are used, the reduced uncertainties in the estimation of $\xi$ allow further investigation of the dependence of $\xi$ on duration. We find that $\xi$ decreases with duration when taking the average of the investigated stations in Germany. We believe that this finding provides a good basis to explore a potentially more suitable formulation of $\xi(d)$ in future studies.

**References**

- Koutsoyiannis, D., Kozonis, D., and Manetas, A.: A mathematical framework for studying rainfall intensity-duration-frequency relationships, J. Hydrol., 206, 118–135, https://doi.org/10.1016/S0022-1694(98)00097-3, 1998.
- Lehmann, E., Phatak, A., Soltyk, S., Chia, J., Lau, R., and Palmer, M.: Bayesian hierarchical modelling of rainfall extremes, in: 20th International Congress on Modelling and Simulation, Adelaide, Australia, edited by Piantadosi, J and Anderssen, RS and Boland, J, pp. 2806–2812, 2013.

- Van de Vyver, H.: Bayesian estimation of rainfall intensity–duration–frequency relationships, J. Hydrol., 529, 1451–1463, https://doi.org/10.1016/j.jhydrol.2015.08.036, 2015. Van de Vyver, H.: A multiscaling-based intensity–duration–frequency model for extreme precipitation, Hydrol. Process., 32, 1635–1647, https://doi.org/10.1002/hyp.11516, 2018.
- Van de Vyver, H. and Demarée, G. R.: Construction of Intensity–Duration–Frequency (IDF) curves for precipitation at Lubumbashi, Congo, under the hypothesis of inadequate data, Hydrolog. Sci. J., 55, 555–564, https://doi.org/10.1080/02626661003747390, 2010.

**Specific comments**

**l 4:**

"IDF curves are steeper": I think this is not understandable in the abstract

**Answer:**   We suggest to rephrase the sentence to: "The monthly IDF curves of the summer months exhibit a more rapid decrease of intensity with duration, as well as higher intensities for short durations than the IDF curves for the remaining months of the year."

**l 5:**

"short convective events occur very likely in summer" : the "very likely" may be confusing because it is stille rare (probability vs conditional probability)

**Answer:**   We suggest to change the sentence to: "Thus, when short convective extreme events occur, they are very likely to occur in summer everywhere in Germany."

**l 24:**

Kuntz et al is in german so I could not check

**Answer:**   The book "Klimawandel in Deutschland" (Climate Change in Germany) summarizes results from a wide range of studies on the observed and projected effects of climate change in Germany. Chapter 7 discusses changes in precipitation. Unfortunately, there is no English edition. Non-German speaking readers are referred to Moberg and Jones (2005) and Łupikasza (2017).

**l 95:**

you consider both 1 and 2 minutes. As shown later in your figures, the distributions for 1 and 2 minutes are very similar so I'm not sure that the 2 minute is necessary.

**Answer:**   We chose to cover a wide range of durations, with the aim of having equidistant durations on the logarithmic scale. Therefore, both one minute and two minutes are included in the analysis. Although the distributions have very similar characteristics, we do not consider removing either of these information as beneficial. Especially since the information about the short durations is crucial for the estimation of the duration offset parameter $\theta$.

**l 178:**

Actually to the best of my understanding, Jurado et al 2020 conclude that accounting for dependence gives better results when d<=10h, which is the case for 10/14 (71%) of the considered durations. So I'm not convinced by your justification (but I agree that accounting for dependence increases much model complexity)

**Answer:** You are correct that Jurado et al. (2020) have shown that the estimation of the higher quantiles for short durations improves when dependence is taken into account. However, this improvement is rather modest considering the increased complexity of the model. In addition, sub-hourly durations were not included in the study. We suggest rewording the sentence as follows: "Jurado et al. (2020) have shown that accounting for asymptotic dependence between durations yields a modest improvement in the estimation of quantiles of short durations $d \leq 10\,\mathrm{h}$, but comes at the cost of increased model complexity. We therefore decide to neglect the dependence between durations when estimating the d-GEV parameters using equation (12)."

**l 205-206:**

I'm not familiar with the cross-validated likelihood method so I missed this part. E.g. what is the number of folds? Please consider being more specific here.

**Answer:** Thank you for pointing this out. We will elaborate more on this part.

**l 233-238:**

to be sure: do you use the same sampling years also for deriving the annual GEV from the monthly GEV ? (i.e. do all monthly GEV use the same sampling years?)

**Answer:** Yes, all monthly maxima (for all durations) from a particular year are jointly sampled.

**l 297-303:**

If I understood correctly, the sum of the ordinates of the dots of a given duration is equal to 0.1 (due to the 0.9-quantile). So dividing the ordinates by 0.1 gives the proportion of exceedances that occur a given month. Wouldn't it be easier to interpret Fig 4 this way? For example in Bever 45% of the exceedances occur in July.

**Answer:** Thank you for this helpful comment. Yes, the product of the monthly probabilities is 0.1, so when dividing by 0.1 we obtain the conditional probabilities. We will include this as additional information in the text and caption.

**Fig 5:**

I don't understand the legend for Pr max. For example what do you mean by "$< 2\%$"? Isn't it "$1 - 2\%$"? Also I would find it easier to interpret if you divide Pr by 0.1, as said above.

**Answer:**

You are right, we will change the legend to $1 - 2\%$. We will mention the information about the conditional probabilities in the caption.

**l 364-366**

I don't understand how you deal for cases p>1-1/T. What is the observed quantile in this case? I guess you consider the maxximul value by I don't thin that's correct. So I suggest removing the cases without dots in Fig 6.

**Answer:** The Quantile Score is defined as

$$\mathrm{QS}(p) = \frac{1}{N} \sum_{n=1}^{N} \rho_p(o_n - q_p), \quad \text{where } \rho_p(u) = \begin{cases} pu & , u \geq 0 \\ (p-1)u & , u < 0, \end{cases}$$

where $u = o_n - q_p$ is the difference between an observation $o_n$ and the quantile $q_p$ estimated from (in this case) the d-GEV model. Therefore the model is always compared directly to all data and no empirical

quantile estimation is necessary. A common problem in the verification of extreme value models is the lack of data to verify the model for high non-exceedance probabilities. However, even if no observations are above the estimated quantile, QS still provides an estimate of the model performance. We decided to indicate the length of the time series available for verification, since the estimate might be less reliable if the length of the time series $T$ is shorter than the period corresponding to the non-exeedance probability $p = 1 - 1/T$ being verified. One should therefore be more cautios when interpreting these results. However, we do not agree that this information should be completely removed.

**l 426:**

I agree that Fisher information matrix is correct in this case but for comparison puropose, I suggest using a bootstrap method as for the other cases.

**Answer:** Thank you for pointing this out. You are correct that it is better for reasons of comparability to estimate all uncertainties using the same method. We will change this and estimate all uncertainties via the boostrap method in Figure 7.

**Fig 8:**

As said above, the monthly GEV seems log-linearly decreasing with duration

**Answer:** See above.

**Section 4:**

As said above, this is actually almost only an extended summary. Please consider shortening it.

**Answer:** See above.

**l 562-563:**

As a first try, I wouldn't try the multi scaling model but I'd rather consider xi function of d.

**Answer:** We agree. We were just suggesting multiscaling could lead to similar results. We will make it more clear in the manuscript, that future studies should investigate the dependence of $\xi$ on duration.

---

## Author Comment (AC3)

**Reply on RC3**

Jana Ulrich, Felix S. Fauer and Henning W. Rust

August 26, 2021

Dear Reviewer,

Thank you very much for your detailed and constructive comments. We address them point by point below.

**Main comments**

**1.**

While the results are informative and supported by extensive analysis, the motivation for this very detailed investigation was not clear from the introduction. IDF curves are typically used for design of long-lived infrastructure systems where monthly variations are essentially irrelevant. The authors discuss a few reasons that monthly IDF curves could be valuable, including to support agricultural or water resources stakeholders. However, these stakeholders typically care about monthly average rainfall, and there is no reasoning in the paper that supports why they would be interested in extremes. Could the authors elaborate and/or find a reference that supports this?

**Answer:**  Thank you for pointing out that the motivation might not be clearly stated. We will revise the introduction to communicate this more clearly.

In addition to the availability of more data and the resulting reduction in uncertainty when estimating annual IDF curves, there are three further aspects that suggest that studying intra-annual variations in the IDF relationship is relevant.

- For stakeholders who use IDF curves for water management rather than planning of infrastructure, the additional information on intra-annual variations may be beneficial. An important example is the recent extreme precipitation event in the states of North Rhine-Westphalia and Rhineland-Palatinate in western Germany. The event also affected the Bever dam, where the example station Bever-Talsperre of this study is located. The intense long-lasting rainfall on 7/14/2021 caused the Bever dam to spill in a controlled manner. There was concern that the resulting higher water level in the Beverteich lake below could cause the dam there to breach, with serious consequences for the downstream villages. As a result, the residents were evacuated. We therefore think that both the information about the time scale and the seasonality of extreme precipitation is of importance for stakeholders who, for example, manage the water level of reservoirs.
- The study of seasonal variations allows drawing new conclusions about the underlying processes, since the use of annual maxima of different durations does not take into account that these may originate from different seasons.
- Changes in the frequency and intensity of extreme precipitation in Europe have been found to differ between different seasons (e.g. Moberg and Jones, 2005; Łupikasza, 2017). Modeling the seasonal variations of extreme precipitation on different time scales is a first important step to detect and interpret the changes in seasonality in a consistent way.

**References**

- Moberg, A. and Jones, P. D.: Trends in indices for extremes in daily temperature and precipitation in central and western Europe, 1901–99, Int. J. Climatol., 25, 1149–1171, https://doi.org/10.1002/joc.1163, 2005.
- Łupikasza, E. B.: Seasonal patterns and consistency of extreme precipitation trends in Europe, December 1950 to February 2008, Clim. Res., 72, 217–237, https://doi.org/10.3354/cr01467, 2017.

**2.**

It seems instead that the motivation for the paper is to "increase understanding" of monthly extremes and examine "underlying mechanisms." If so, then what new information do monthly IDF curves bring? Is this simply a convenient way to evaluate monthly extremes and also account for storm duration? Or could monthly IDF curves bring added value to engineering analyses?

**Answer:** Two major new insights we found using monthly IDF curves are:

- It could be incorrect to assume a simple scaling of intensity with duration for durations $\geq 1\,\text{h}$, especially at stations with a prominent shift in seasonality from short to long durations. We could show that such a shift can lead to a decreased slope of the IDF curves for long durations $\geq 24\,\text{h}$. While it is also possible to model this deviation by adjusting the IDF model for annual maxima, this would not enable us to understand its cause.

- Due to the large uncertainties associated with the estimation of the shape parameter $\xi$, it has so far not been possible to derive an empirical relationship between $\xi$ and duration. We were able to show that a more reliable estimate of this parameter is possible when seasonal variations are incorporated. On average, $\xi$ decreases with duration across all investigated stations. This interesting result should be investigated in more detail in future studies.

**3.**

The introduction does mention that monthly IDF curves could bring added value compared to annual block maxima by including more data in the analysis. However, there is an existing technique, called "peaks over threshold" (POT), that evaluates all storms in a year over a certain threshold. It is unclear whether the monthly maximum technique brings added value compared to the POT method, but it is clear that the monthly maxima method is not the only way to include more data in the analysis. There are drawbacks to POT, of course, including that the annual return period is no longer directly interpretable since more than one storm per year can be included in the extreme event series. But the POT technique should be mentioned in the introduction as an alternative way to include more data. A comparison of the monthly maximum technique to POT should also be mentioned in the conclusions/future work section.

**Answer:** We agree that the POT approach is an alternative method for modeling precipitation extremes. We will mention this in the introduction. However, the POT approach alone does not allow to study seasonal variations within the IDF relationship. To do so, one would need a seasonally varying threshold. Within a duration-dependent distribution, the threshold would additionally depend on the duration. We consider this approach to be much more complex for the problem at hand and consider the d-GEV approach as a more suitable choice in this context. We this motivation for our choice to the introduction.

**4.**

Based on these points, a distinct motivation for the creating of monthly IDF curves seems to be missing. After reading the results, it seems that monthly IDF curves could bring some added value in terms of uncertainty evaluation and potentially even for parameter fitting. This is of interest to an engineering audience who are developing and using IDF curves. Suggest restructuring the introduction to ensure the typical IDF curve audience understands this before reading the entire, very detailed study.

**Answer:** Thank you for pointing this out. We will change the introduction to more clearly communicate the benefits mentioned above.

**5.**

The motivation or added value for creating monthly IDF curves could also be discussed further (in results or discussion section). Is it worth it to use monthly maximum instead of annual maxima? If so, in which cases: for annual IDF curves in general, or only when we are interested in monthly extremes? Why?

**Answer:** The advantages of modeling monthly maxima are:

- reduced uncertainties in parameter and quantile estimation due to more available data points
- seasonal information
- better understanding of the processes/ duration dependence of the parameters.

They come at the cost of a more complex model. It is therefore suitable to use the monthly maxima when there are large differences in the seasonality of extreme events on different time scales, such as at the station Bever-talsperre, or for stations where only short observation time series are available. However, it must be considered that a misspecification of the seasonal variations of the parameters can lead to poor results. Future studies should therefore investigate whether the assumptions made for the intra-annual variations of the d-GEV parameters can be relaxed further. Moreover, modeling monthly precipitation maxima with the GEV may not be possible in regions with very small precipitation amounts during some months of the year. Therefore, the applicability of the model to the data should always be verified.

We will add this information to the discussion.

**6.**

The discussion section, which repeats a lot of the results, could be condensed, or merged with the results section.

**Answer:** We agree and will combine the discussion section with the results section.

**Specific comments**

**Line 25 and line 32**

Similar to general comment. Why it is "critical" to provide information about extremes on a monthly basis?

**Answer:** See above.

**Line 47 − 52**

This is a common problem, not just in Germany. Many places (like the US, NOAA NCDC) have 50 years or more of daily data, with data at sub-hourly resolutions only available in the past decade or so. I suggest making this statement more generalizable, and say this is also the case for Germany, the focus of this study. Many others will also be interested in using the available data more efficiently through pooling information.

**Answer:** We will change this accordingly.

**Line 59 − 64**

Yes, block maxima typically are only used for annual maximum because other methods like peaks over threshold (see Coles) are used if you want to capture more data and extremes within a year. Why not use the peaks over threshold method instead of monthly block sizes? If monthly variation is relevant, why are periodic functions needed as covariates (instead of a GEV distribution dependent on duration and month)? It

seems that later on you clarify this – parameters can be reduced. Suggest clarifying this in the introduction as well and that you will compare the two techniques later on.

**Answer:** We will mention in the introduction, that the POT approach is an alternative method for modeling precipitation extremes, which can make more use of the available data than the block-maxima approach. We consider it more complex, however, to include seasonal variations into a duration-dependent GPD distribution.

We will also mention that smooth periodic functions as covariates are used to reduce the number of parameters which need to be estimated in the introduction.

**Line 65**

Did Fischer et al compare this method to peaks over threshold? More precise quantile estimates compared to what?

**Answer:** Fischer et al. (2018) compared the annual return levels of 24 h precipitation sums estimated from a GEV using annual maxima to those estimated when modeling monthly maxima using a GEV with smooth periodic functions as covariates.

**Line 74, research question 3**

this question is unclear and should be briefly introduced in the introduction.

**Answer:** Thank you for pointing this out, we will add an explanation in the introduction.

**Line 137 − 138**

unclear what is meant by "identically distributed precipitation cannot be motivated if an annual cycle exists..." Please clarify. Is it an interannual cycle or intra-annual? Also, do you mean independent identically distributed?

**Answer:** A basic assumption of the Fisher-Tippett-Gnedenko theorem is that the random variables within each block are independent and identically distributed (iid). However, we can see that there is a clear intra-annual cycle for the precipitation intensity and this assumption is therefore violated when modeling annual maxima. We will reword this sentence to make it more clear.

**Line 139**

what is meant by "sufficient"? Meaning it can be used? It is a compromise? Also, this sentence is repeated from the introduction. Suggest rewording, shortening, or removing.

**Answer:** We will reword this sentence. Briefly, sufficient in this context means that the GEV distribution is well suited for the description of the monthly maxima, despite the smaller block size. For a more detailed explanation, please see our response to Reviewer Comment 1 https://hess.copernicus.org/preprints/hess-2021-336/hess-2021-336-AC1-supplement.pdf.

**Line 215**

choice to keep the shape and theta parameters constant is justified. How so? It seems these parameters are varying in the same fashion as mu, the modified location parameter? A bit more explanation here would be useful.

**Answer:** The choice to keep the shape parameter $\xi$ and the duration offset parameter $\theta$ constant is discussed in the manuscript in lines 187-197. We agree that we have not sufficiently discussed our choice of a varying modified location parameter $\tilde{\mu}$. We will add our reasoning to the manuscript.

The modified location parameter is defined as

$$\tilde{\mu} = \mu(d)/\sigma(d).$$

Therefore setting $\tilde{\mu}(\text{doy}) = \text{const.}$ would enforce the annual cycle of the location parameter $\mu$ and the scale parameter $\sigma$ to be in phase for any fixed duration $d$. Maraun et al. (2009) investigated this relationship for daily precipitation sums in the UK and found that this assumption is not justified, because the annual cycles of these two parameters are slightly out of phase. We investigated this in an explorative analysis by modeling the individual durations. For this purpose, we modeled the monthly maxima of each duration using a GEV with monthly covariates. Figure 1 shows the resultig parameter estimates $\mu$ and $\sigma$ for some durations at station Bever-Talsperre in the first two columns as lines. For comparison, the estimates resulting from separately modeling the maxima of each duration and month with a GEV are shown as dots. The right column presents the resulting estimates for $\mu/\sigma$ for each duration. Since $\mu/\sigma$ shows a clear variation for each duration, it can be concluded that the assumption of phase equality of $\mu$ and $\sigma$ for each duration may be too restrictive. Based on this exploratory analysis, we decided to adopt a varying parameter $\tilde{\mu}$ throughout the year.

[Figure]

Figure 1: Parameter estimates for fixed durations modeling monthly maxima using (1) a GEV with monthly covariates (purple lines) and (2) a separate GEV model for each month (dots). The error bars and shaded areas show the 95% confidence intervals obtained via the estimated Fisher information matrix.

We suggest to add the additional information about the exploratory analysis to the appendix.

**Reference**

- Maraun, D., Rust, H. W., and Osborn, T. J.: The annual cycle of heavy precipitation across the United Kingdom: a model based on extreme value statistics, Int. J. of Climatol., 29, 1731–1744, https://doi.org/10.1002/joc.1811, 2009.

**Line 350**

could you comment on what this implies for IDF created with annual block sizes? Does it matter?

**Answer:**  It does not seem to affect the quantiles estimated from the annual maxima, at least at stations where the annual maxima of different durations are from approximately the same seasons, such as stations (3)-(6) in Figure 4 in the manuscript. When a stronger shift in seasonality between short and long durations exists, as at station (1) and (2), the IDF model as defined in equations (4)-(7) in the manuscript is not able to reproduce this shift. This can be observed in Figures 6 (left) and B1 in the manuscript. The annual maxima of the individual durations originate from different seasons or distributions and therefore do not follow a simple power-law for longer durations. However, this could be resolved by adjusting the annual IDF model.

**Lines 352 − 354**

The authors state that the annual maxima originate from effective blocks of different sizes, seasons, etc. Could you comment more on why this is a problem when annual block sizes are used? Wouldn't the annual maxima still be captured? Does it matter when it occurred?

**Answer:**  It does not matter as long as the effects caused by a shift in seasonality in the annual maxima of different durations are accounted for in the annual IDF model. To accomplish this, these effects must first be understood. Therefore modeling of the seasonality is a crucial step.

---

## Author Response (AR2)

**Author's response**

Jana Ulrich, Felix S. Fauer and Henning W. Rust

November 9, 2021

**Dear Nadav Peleg,**

we are pleased that you have accepted our manuscript. As required, we have made the following changes:

- We transferred the contents of the Appendix to a separate file as a Supplement.

- We updated the reference Fauer et al. (2021).

- We replaced the footnotes with direct references to the sources in the text.

We were advised that the color scale in Figures 3 and S2 might not be color blind conform. However, we already adjusted the color scale accordingly by changing the saturation in addition to the hue. A check of the color scale indicated that the colors are distinguishable even for color blind people.

In addition to the requested changes, we adapted Equation 4 to match the text, since the equation was formulated for the exceedance probability 1-p, but the text states the non-exceedance probability.

Sincerely,

Jana Ulrich (on behalf of all authors)